# Impaired prosaposin lysosomal trafficking in frontotemporal lobar degeneration due to progranulin mutations

Xiaolai Zhou[1], Lirong Sun[1,2], Oliver Bracko[3], Ji Whae Choi[1], Yan Jia[1], Alissa L. Nana[4], Owen Adam Brady[1], Jean C. Cruz Hernandez[3], Nozomi Nishimura[3], William W. Seeley[4,5] & Fenghua Hu[1]

Haploinsufficiency of progranulin (PGRN) due to mutations in the granulin (*GRN*) gene causes frontotemporal lobar degeneration (FTLD), and complete loss of PGRN leads to a lysosomal storage disorder, neuronal ceroid lipofuscinosis (NCL). Accumulating evidence suggests that PGRN is essential for proper lysosomal function, but the precise mechanisms involved are not known. Here, we show that PGRN facilitates neuronal uptake and lysosomal delivery of prosaposin (PSAP), the precursor of saposin peptides that are essential for lysosomal glycosphingolipid degradation. We found reduced levels of PSAP in neurons both in mice deficient in PGRN and in human samples from FTLD patients due to *GRN* mutations. Furthermore, mice with reduced PSAP expression demonstrated FTLD-like pathology and behavioural changes. Thus, our data demonstrate a role of PGRN in PSAP lysosomal trafficking and suggest that impaired lysosomal trafficking of PSAP is an underlying disease mechanism for NCL and FTLD due to *GRN* mutations.

[1] Department of Molecular Biology and Genetics, Weill Institute for Cell and Molecular Biology, Cornell University, Ithaca, New York 14853, USA. [2] Department of Neurobiology, School of Basic Medical Sciences, Southern Medical University, Guangzhou 510515, China. [3] Nancy E. and Peter C. Meinig School of Biomedical Engineering, Cornell University, Ithaca, New York 14853, USA. [4] Department of Neurology, University of California, San Francisco, California 94158, USA. [5] Department of Pathology, University of California, San Francisco, California 94158, USA. Correspondence and requests for materials should be addressed to F.H. (email: fh87@cornell.edu).

Frontotemporal lobar degeneration (FTLD) is the most prevalent form of early-onset dementia after Alzheimer's disease (AD), and accounts for 20–25% of presenile dementias[1]. A large subset of individuals with FTLD have evidence of ubiquitin-positive inclusions comprised of the protein TDP-43 (referred to as FTLD-TDP)[2,3]. Haploinsufficiency of the granulin (GRN) gene is one of the major causes of FTLD-TDP[4–6]. GRN encodes an evolutionarily conserved, secreted glycoprotein of 7.5 granulin repeats (progranulin, PGRN), and its function in the nervous system is still not well understood[7–10]. Emerging evidence suggests a role for PGRN in regulating lysosomal function. First, individuals who are homozygous for mutant GRN exhibit neuronal ceroid lipofuscinosis (NCL)[11,12], which is a group of lysosomal storage diseases characterized by the accumulation of autofluorescent storage material (lipofuscin) in neurons and other cell types[13,14]. Mice lacking PGRN also accumulate lipofuscin[15] and share several common pathologies with animal models of other NCL disease[16]. Second, GRN is transcriptionally coregulated with a number of essential lysosomal genes by the transcriptional factor TFEB[17]. Third, within cells, PGRN is localized to lysosomes and two independent PGRN lysosomal trafficking pathways have been identified[18,19]. However, the precise lysosomal function of PGRN remains unclear.

Accumulating evidence also suggests that lysosomal dysfunction might serve as a common mechanism in NCL and FTLD-TDP associated with GRN mutations. A recent study has shown that FTLD patients with GRN mutations also exhibit typical pathological features of individuals with NCL, including accumulations of saposin D and subunit c of mitochondrial ATP synthase (SCMAS)[20]. Additionally, mice deficient in Ctsd, one of the NCL genes, develop TDP-43 aggregates[20], a hallmark of FTLD-TDP, supporting the idea that FTLD and NCL are pathologically linked. Therefore, understanding how PGRN regulates lysosomal functions may be helpful in understanding the shared molecular mechanisms underlying FTLD and NCL with GRN mutations.

Here, we report that PGRN facilitates lysosomal trafficking of prosaposin (PSAP), the precursor of lysosomal saposin activators essential for glycosphingolipid degradation in the lysosome[21–23], and determines the levels of saposins in neurons. Reduced PSAP and saposin levels are observed in neurons, both in mice lacking PGRN and in samples from humans haploinsufficient for PGRN. Furthermore, reduction in PSAP levels leads to FTLD-like phenotypes in mice, supporting the idea that impaired lysosomal trafficking of PSAP might be a shared disease mechanism in FTLD and NCL caused by GRN mutations.

## Results

### PGRN bridges the interaction between PSAP and sortilin.
In a proteomic screen searching for PGRN-binding partners[19], we uncovered a novel interaction between PGRN and PSAP, the precursor of saposin peptides (A, B, C and D) that are essential for glycosphingolipid metabolism in the lysosome[21–23]. PSAP or saposin deficiency is known to cause several distinct lysosomal storage disorders, including Gaucher disease, Krabbe disease and metachromatic leukodystrophy[21–23]. Lysosomal PSAP and saposins can be derived from the biosynthetic pathway (sorting at trans-Golgi network) or from the extracellular space via the endocytic pathway since PSAP is a secreted glycoprotein. Several receptors have been shown to mediate PSAP lysosomal trafficking, including the cation-independent mannose 6-phosphate receptor (M6PR)[24,25], sortilin[26] and LRP1[27]. Since both PGRN and PSAP have been reported to bind to sortilin[18,26], we investigated whether PSAP and PGRN compete with each other

for sortilin binding. However, in contrast to published data[26], we failed to detect an interaction between PSAP and sortilin by co-immunoprecipitation (IP) even when both proteins are overexpressed in HEK293T cells (Fig. 1a,b) or in the COS-7 cell surface-binding assay with alkaline phosphatase (AP)-tagged PSAP (Fig. 1c). Interestingly, PSAP and sortilin strongly associate with each other in the presence of PGRN in the co-IP assay (Fig. 1a,b) and in the COS-7 cell surface-binding assay (Fig. 1c), suggesting that PGRN bridges the interaction between PSAP and sortilin.

### PGRN facilitates PSAP lysosomal delivery via sortilin.
Sortilin has been shown to mediate lysosomal delivery of PGRN[18]. Since PGRN mediates the interaction between PSAP and sortilin, we tested whether PGRN facilitates lysosomal trafficking of PSAP through sortilin. Previously, we showed that uptake of PGRN does not occur in COS-7 cells unless sortilin is exogenously expressed[18]. Here we found that in COS-7 cells transfected with sortilin, exogenously added FLAG-tagged PSAP shows no detectable uptake and trafficking to lysosomes (Fig. 1d). However, the presence of recombinant PGRN greatly facilitates PSAP binding to sortilin-expressing COS-7 cells and subsequent lysosomal targeting as shown by colocalization with the lysosomal protease cathepsin D (Fig. 1d). Furthermore, endocytosed PSAP is processed into intermediates and saposin peptides, confirming successful lysosomal delivery (Fig. 1e).

To confirm that PGRN facilitates PSAP lysosomal trafficking with endogenous levels of sortilin expression, we first examined the binding of exogenous PSAP to cultured cortical neurons at DIV12, which are known to express high levels of sortilin[18]. AP-tagged PSAP demonstrated specific binding to both neuronal cell bodies and processes that can be displaced by untagged PSAP (Fig. 2a,b). Binding of PSAP to the neuronal cell surface is greatly enhanced in the presence of recombinant PGRN (Fig. 2a,b), which results in much more PSAP uptake (Fig. 2c–e). To determine whether sortilin is the receptor for PGRN uptake in cortical neurons, we compared the uptake of full-length PGRN compared with PGRNΔ3aa, which has the sortilin-binding site in PGRN deleted[28]. PGRNΔ3aa recombinant protein cannot be taken up by neurons and fails to enhance neuronal PSAP uptake, confirming that sortilin is the primary receptor for PGRN on the neuronal cell surface (Fig. 3a–c). The presence of GST-RAP (glutathione S-transferase-tagged receptor-associated protein), which blocks LRP1 ligand interactions[27], reduces PSAP uptake by ~75% (Fig. 3a, 3c), suggesting that LRP1 is the main receptor for PSAP uptake in cortical neurons. GST-RAP also partially blocks PGRN uptake, in line with previous report that RAP blocks the interaction between sortilin and its ligands[29]. In addition, PSAP enhances PGRN uptake (Fig. 2c–e, Fig. 3a, 3b) and facilitates the uptake of PGRNΔ3aa protein, which is not taken up by neurons on its own (Fig. 3a,b). These data, together with our previous findings[19], strongly support that PGRN and PSAP facilitate each other's uptake and lysosomal trafficking via their respective receptors, sortilin and LRP1.

### PGRN facilitates neuronal uptake of glial-derived PSAP.
To understand the physiological interactions of PGRN and PSAP in vivo, we examined PGRN and PSAP expression patterns in the mouse brain under normal conditions and in disease-like states. In the normal brain, PGRN is expressed in both neurons and microglia (Supplementary Fig. 1), consistent with a previous study[18]. PSAP and the PGRN receptor, sortilin, exhibit a punctate localization in neurons with minimal expression in microglia (Supplementary Fig. 1). Since the levels of PGRN are known to be upregulated in microglia under inflammatory

conditions[18], we also determined the expression of PSAP in response to injury. In response to an acute cortical stab injury, the protein levels of PGRN and, to a lesser extent, PSAP are upregulated (Fig. 4a–c). Costaining with Iba1 showed that PGRN and PSAP are highly expressed in activated microglia in response to injury (Fig. 4d, 4e). Additionally, increased PSAP

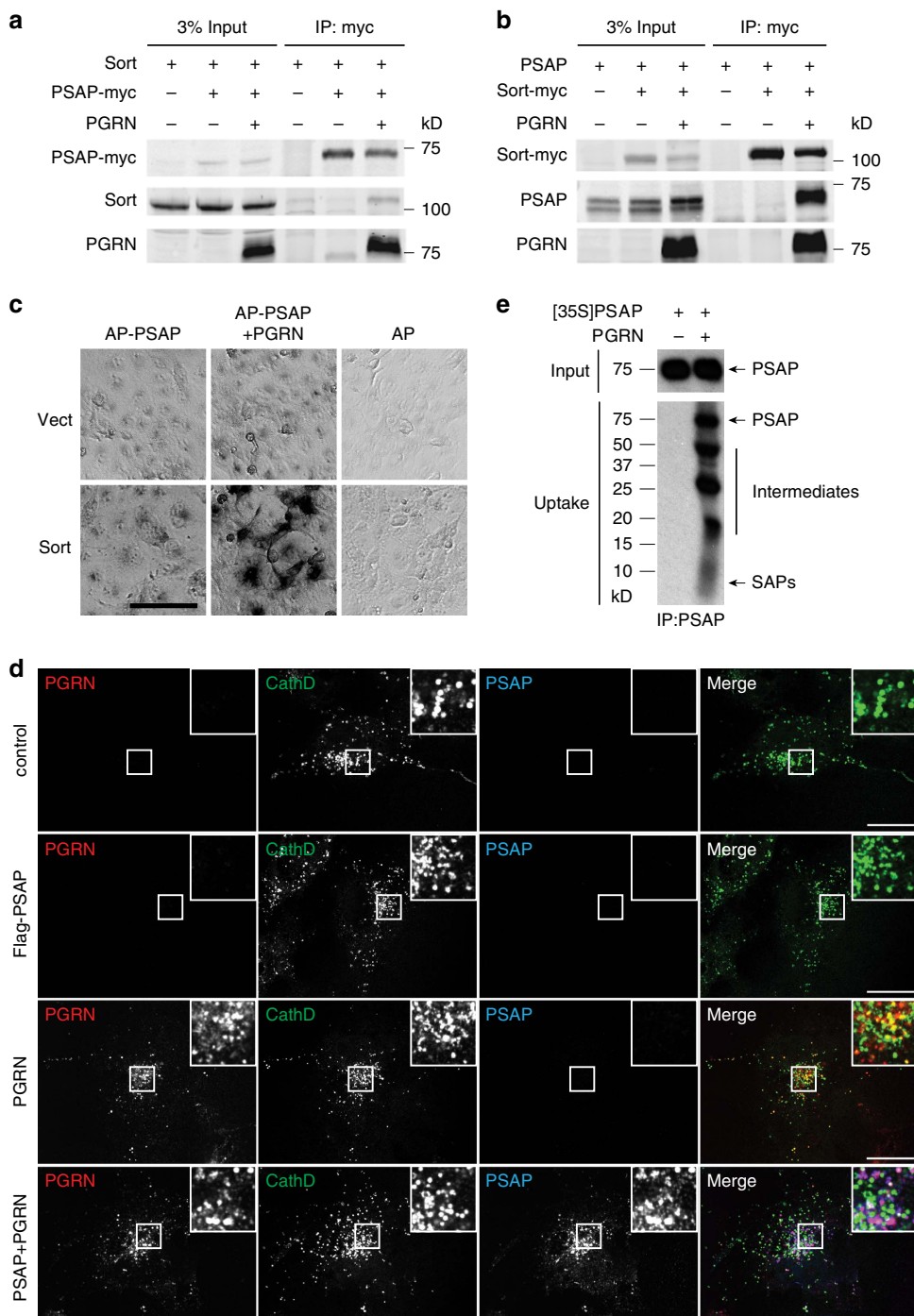

**Figure 1 | PGRN bridges the interaction between PSAP and sortilin and facilitates lysosomal targeting of PSAP via sortilin.** (**a**) Myc-tagged PSAP-, PGRN- and sortilin (Sort)-expressing constructs were transfected into HEK293T cells as indicated. Cell lysates were subject to anti-myc immuno-precipitation and blotted with anti-sortilin, myc and PGRN antibodies. (**b**) PSAP, PGRN and myc-tagged sortilin (Sort)-expressing constructs were transfected into HEK293T cells as indicated. Cell lysates were subject to anti-myc immunoprecipitation and blotted with anti-sortilin, myc and PGRN antibodies. (**c**) COS-7 cells transfected with an empty vector (Vect) or sortilin (Sort)-expressing construct were incubated with AP-tagged PSAP alone or AP-PSAP with PGRN. Scale bar, 100 μm. (**d**) Sortilin-expressing COS-7 cells were treated with recombinant FLAG-PSAP (1 μg ml$^{-1}$) and/or his-PGRN (1 μg ml$^{-1}$) as indicated at 37 °C for 5 h. Cells were costained with anti-FLAG, anti-PGRN and anti-cathepsin D antibodies. Scale bar, 20 μm. (**e**) Sortilin-expressing COS-7 cells were incubated with radiolabelled CM containing PSAP with or without recombinant his-PGRN (1 μg ml$^{-1}$) for 24 h before lysis and immunoprecipitation with anti-PSAP antibodies. The immunoprecipitation products were separated on tricine gels and visualized by radiography. (**a**–**e**) The representative images from three independent experiments.

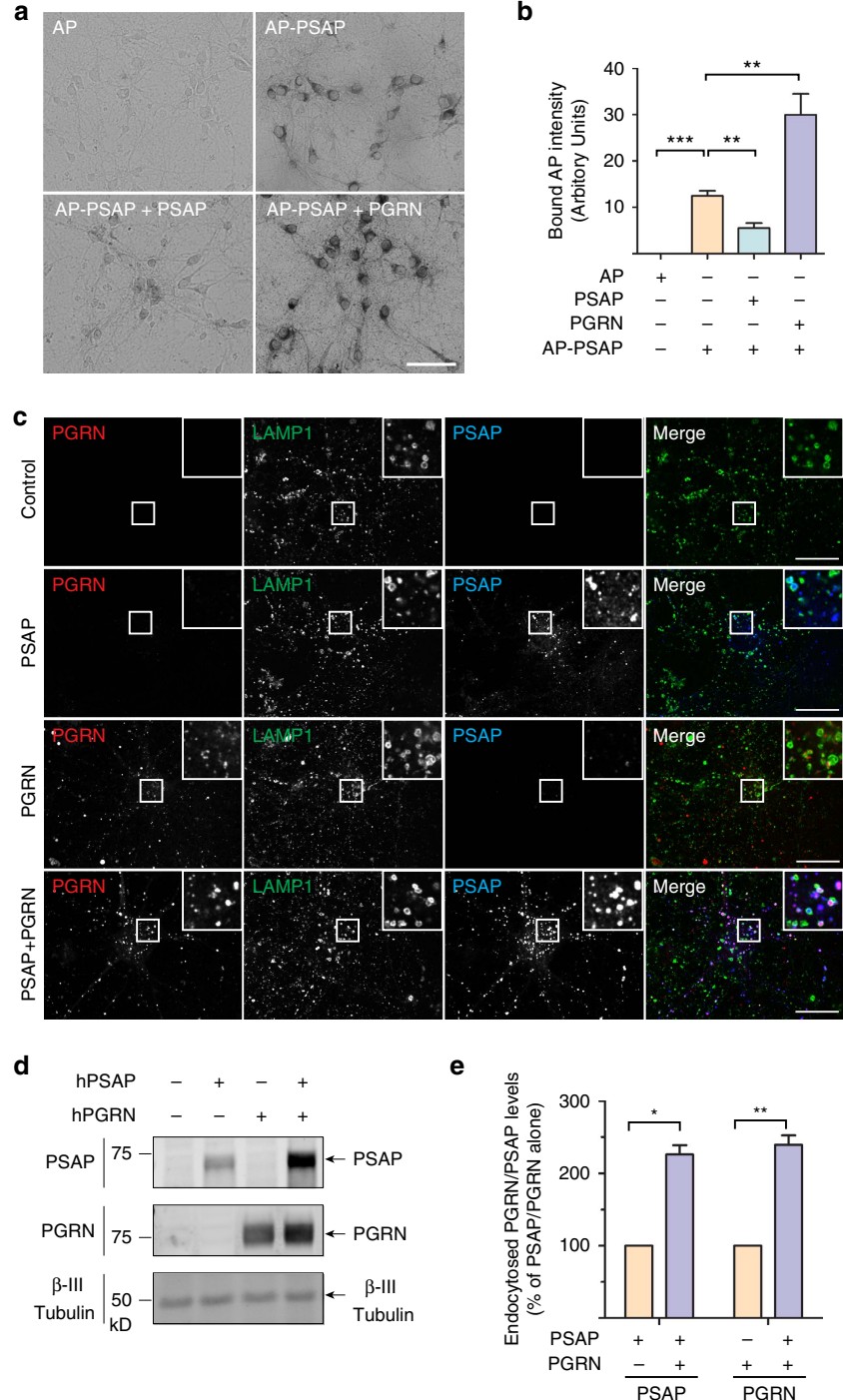

**Figure 2 | PGRN facilitates uptake and lysosomal targeting of PSAP in primary cortical neurons.** (**a**) Primary cortical neurons (DIV12) were incubated with AP-PSAP (50 nM) alone, or together with purified recombinant his-PSAP (10 μg ml$^{-1}$), or his-PGRN (1 μg ml$^{-1}$) as indicated. Scale bar, 50 μm. (**b**) Quantification of bound AP intensity of (**a**); $n = 3$, ***$P < 0.001$, **$P < 0.01$, one-way analysis of variance (ANOVA). Data are presented as mean ± SEM. (**c**) Primary cortical neurons (DIV12) were treated with recombinant hPSAP (1 μg ml$^{-1}$) and/or hPGRN (1 μg ml$^{-1}$) for 16 h as indicated. Cells were stained with anti-mouse LAMP1, anti-human saposin B and anti-human PGRN antibodies. Scale bar, 20 μm. Representative images from three independent experiments were shown. (**d**) Primary cortical neurons (DIV12) were treated as in **c**. The cells were collected and subjected to immunoblotting with anti-human saposin B, anti-human PGRN and anti-β III tubulin antibodies. (**e**) Quantification of endocytosed neuronal PSAP and PGRN in (**d**), normalized to PSAP or PGRN alone. $n = 3$, **, $p < 0.001$, *, $p < 0.05$, paired $t$-test. Data presented as mean ± SEM. ***$P < 0.001$, Student's $t$-test.

levels are also detected in activated astrocytes (Fig. 4e). Furthermore, the expression of PGRN and PSAP is increased in activated glia during normal ageing (Supplementary Fig. 2). Consistent with these *in vivo* data, western blot analysis with cultured cortical neurons and microglia showed high levels of PGRN and PSAP proteins expressed and secreted from primary microglia and high levels of sortilin in neuronal cell lysate (Fig. 5a–c).

The expression pattern of PGRN, PSAP and sortilin suggests that neuronal sortilin mediates endocytosis of the glia-derived

PGRN–PSAP complex. To determine the role of microglial PGRN in delivering PSAP to neurons, we collected conditioned medium (CM) from radiolabelled microglia from wild-type (WT) mice and mice lacking PGRN and applied it to DIV12 primary

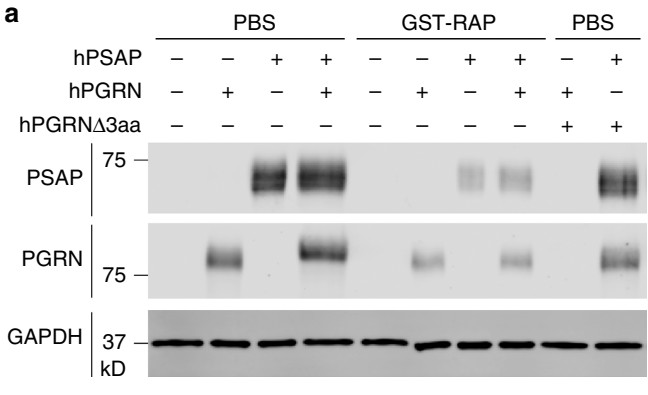

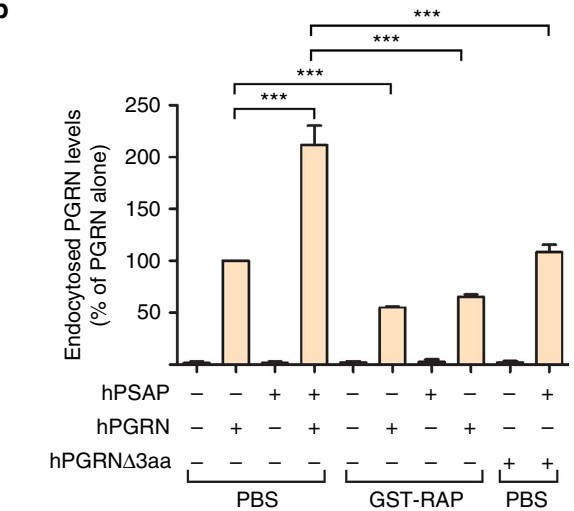

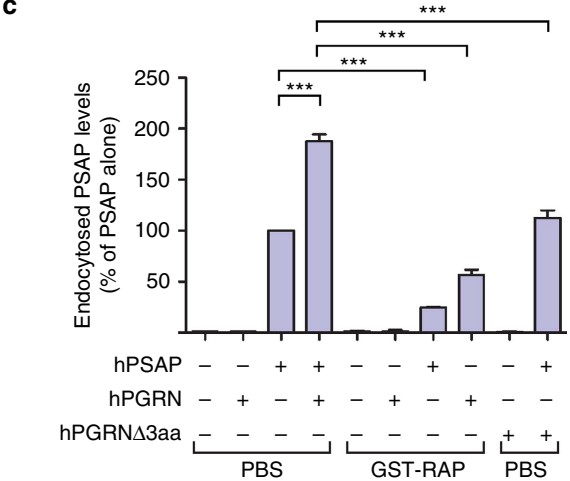

**Figure 3 | PGRN–sortilin interaction enhances PSAP uptake in primary cortical neurons.** (**a**) Primary cortical neurons (DIV12) were pretreated with either GST-RAP (50 μg ml$^{-1}$) or PBS (control) for 30 min, and were then treated with recombinant hPSAP (1 μg ml$^{-1}$) and/or hPGRN (1 μg ml$^{-1}$), hPGRNΔ3aa (1 μg ml$^{-1}$) in the presence or absence of GST-RAP (50 μg ml$^{-1}$) for 16 h as indicated. (**b**) Quantification of endocytosed neuronal PGRN in (**a**), normalized to PGRN alone (set as 100%). $n = 3$, \*\*\*, $P < 0.001$, Repeated Measures ANOVA. Data are presented as mean ± SEM.

cortical neurons (Fig. 5d). Application of conditioned media from WT microglia to neurons leads to much more efficient PSAP uptake and processing than conditioned media derived from $Grn^{-/-}$ microglia 67.7% ±6.0 SEM relative to WT control, although PSAP levels are comparable in the CM of WT versus $Grn^{-/-}$ microglia (Fig. 5e), suggesting that microglia-derived PGRN facilitates neuronal uptake of PSAP. Thus, despite the presence of PSAP receptors M6PR and LRP1 in neurons (Fig. 5c), PGRN enhances PSAP neuronal uptake via sortilin-mediated endocytosis, possibly due to higher levels of sortilin on the neuronal cell surface compared with M6PR and LRP1.

To investigate the role of PGRN in regulating PSAP trafficking *in vivo*, we measured neuronal PSAP levels in adult neurons of WT and $Grn^{-/-}$ mice. Neuronal PSAP signal in neurons is significantly reduced in $Grn^{-/-}$ mice (Fig. 6a,b), despite an increase of PSAP levels in glial cells in $Grn^{-/-}$ mice (Supplementary Fig. 3). Similar results were seen when lysosomal PSAP signals were quantified in WT and $Grn^{-/-}$ neurons (Supplementary Fig. 4). However, PGRN does not appear to have a significant role in PSAP lysosomal trafficking in the biosynthetic pathway as neuroblastoma cells N2a lacking PGRN expression still traffic PSAP to lysosomes (Supplementary Fig. 5). These data together support a role of PGRN in facilitating PSAP lysosomal trafficking from the extracellular space *in vivo*. Consistent with this notion, PGRN deficiency leads to significant increases in serum levels of PSAP protein without changes in PSAP mRNA levels (Fig. 6c,d). Similar results were obtained in sortilin-deficient mice (Fig. 6c,d), supporting an *in vivo* role of PGRN-sortilin in PSAP uptake and lysosomal delivery.

**Neuronal saposin levels are decreased in FTLD-*GRN*.** PGRN haploinsufficiency is a leading cause of FTLD-TDP. To assess whether PSAP trafficking is affected by *GRN* mutations in patients with FTLD-TDP, we stained PSAP and saposins in the orbitofrontal cortex of controls, and in patients with FTLD due to *GRN* mutations (FTLD-*GRN*), or AD. Increased levels of PSAP are detected in the microglia (Fig. 7a,d) and astrocytes (Fig. 7b,e) in tissue from patients with FTLD-*GRN* and AD patients. Despite this increase, a marked reduction of neuronal PSAP (Fig. 7a–c), saposin A (Supplementary Fig. 6a), saposin B (Fig. 8) and saposin C (Supplementary Fig. 6b) signals is observed in FTLD-*GRN* compared with controls or AD, supporting our hypothesis that PGRN mutations lead to reduced neuronal saposin levels in FTLD. To determine whether decreased neuronal saposin levels are specific to FTLD-*GRN*, we also examined the levels of saposin B in more detail in controls, and individuals with AD, FTLD-*GRN* or corticobasal degeneration, a subtype of FTLD with tau inclusions (FTLD-tau). Normal levels of neuronal saposin B were detected in samples from individuals with FTLD-tau (Fig. 8a,c), supporting the idea that decreased neuronal saposin levels in FTLD-*GRN* is not a general feature of all FTLD subtypes. Furthermore, neuronal PGRN levels are directly correlated with neuronal saposin B levels (Fig. 8d), further supporting a critical role of PGRN in determining saposin levels in neurons. Consistent with a previous report[20] and the fact that saposin D is one of the main components of lipofuscin in a vast majority of NCL cases[13], we observed accumulation of neuronal saposin D in FTLD-*GRN* (Supplementary Fig. 7). Careful examination revealed that relatively healthy neurons have lower levels of saposin D (Supplementary Fig. 7a), similar to other saposins but neurons with elevated LAMP1 levels and enlarged lysosomes show increased saposin D intensity (Supplementary Fig. 7b), suggesting that lysosomal dysfunction leads to the accumulation of saposin D. The specific accumulation of saposin D but not other saposins in

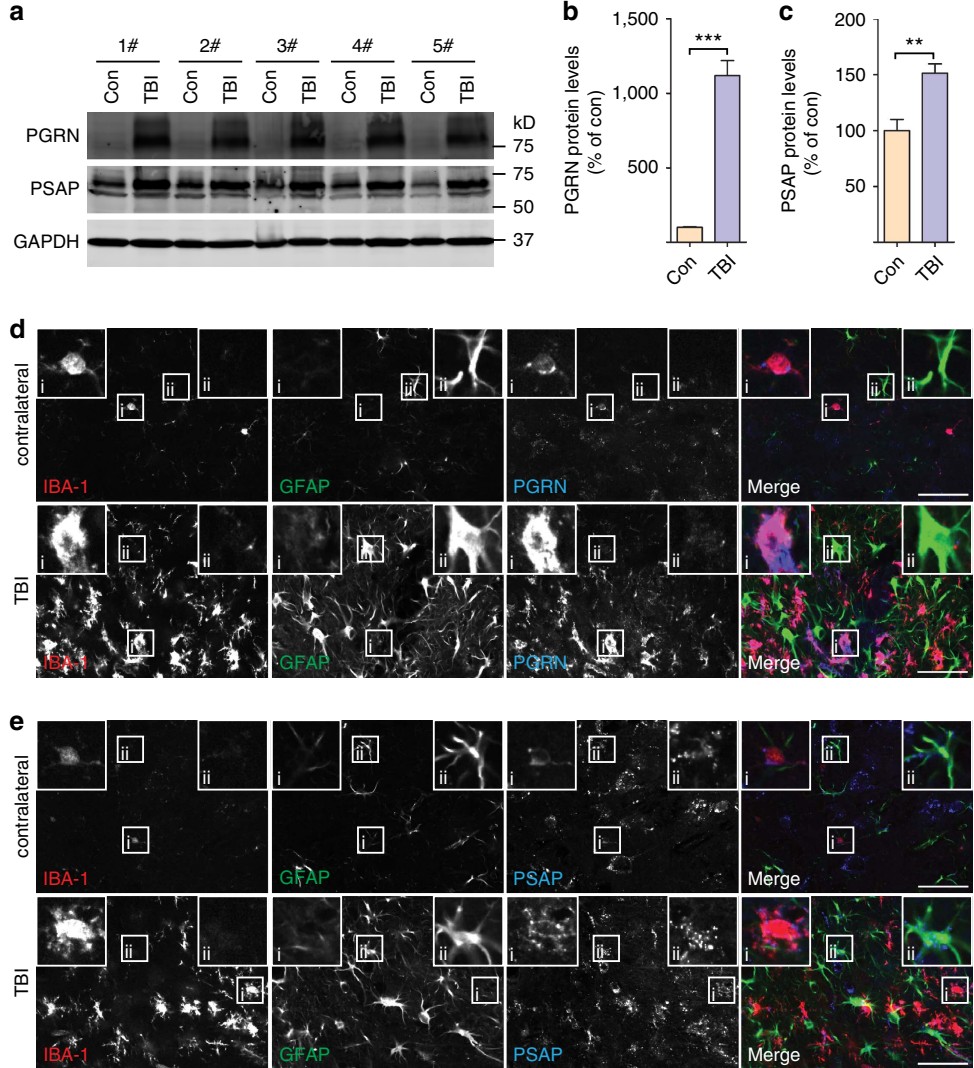

**Figure 4 | Increased levels of PGRN and PSAP in activated glial cells on injury.** (**a**) Brain tissues from 6-month-old WT mouse 4 days after cortical stab wound injury were subjected to immunoblotting with anti-mouse PSAP, anti-mouse PGRN and anti-mouse GAPDH antibodies. TBI, injury hemisphere; Con, conterlateral hemisphere; $n = 5$. (**b**) Quantification of PGRN levels in **a**; $n = 5$, ***$P < 0.001$, Student's $t$-test. Data are presented as mean ± SEM. (**c**) Quantification of PSAP levels in **a**; $n = 5$, **$P < 0.01$, Student's $t$-test. Data are presented as mean ± SEM. (**d**) Brain sections from 6-month-old WT mouse 4 days after cortical stab wound injury were costained with anti-IBA1 (marker for microglia), anti-GFAP (marker for astrocyte) and anti-PGRN antibodies as indicated. Representative image of microglia in shown in inset i and astrocyte in inset ii. Scale bar, 50 μm. (**e**) Brain sections from 6-month-old WT mouse 4 days after cortical stab wound injury were costained with anti-IBA1 (marker for microglia), anti-GFAP (marker for astrocyte) and anti-PSAP antibodies as indicated. Scale bar, 50 μm. Representative image of microglia in shown in inset i and astrocyte in inset ii. (**d,e**) The representative images from two different mice.

FTLD-*GRN* might reflect their different stability and aggregation properties in the lysosome.

**Reduced PSAP function in mice leads to FTLD-like phenotypes.** Since saposin levels are markedly decreased in FTLD-*GRN* patients, we asked whether loss of PSAP function could lead to FTLD-related phenotypes in mice. Because total loss of PSAP leads to death around weaning age in mice, we took advantage of another mouse line ($Psap^{-/-NA}$) that expresses low levels of PSAP ($\sim 5–40\%$ of WT), which can live up to 7 months[30]. Accumulation of p62 and ubiquitin-positive aggregates is evident in these mice compared with age-matched WT controls (Fig. 9a,b). Glial activation is also observed (Fig. 9b). Although we failed to detect TDP-43 aggregation in the $Psap^{-/-NA}$ mice, hyperphosphorylated TDP-43 accumulates in the

insoluble fraction of brain lysates of $Psap^{-/-NA}$ mice (Fig. 9a). The lipophilic SCMAS is another main component of lipofuscin detected in many NCL patients[13], and was found to aggregate in patients with FTLD due to *GRN* mutations[20]. We found that SCMAS accumulates in aged PGRN-deficient mice (Supplementary Fig. 8) and also in the 6-month-old $Psap^{-/-NA}$ mice (Fig. 9b), but not in the age-matched controls. Taken together, these results indicate that partial loss of PSAP could lead to pathological changes in mice similar to those seen in patients with FTLD-*GRN*.

PGRN deficiency in mice leads to FTLD-related behavioural deficits[10,31]. To determine whether loss of PSAP function could lead to FTLD-related phenotypes, we assessed the behaviour of aged $Psap^{+/-}$ mice. In the open-field test, 12-month-old $Psap^{+/-}$ mice show reduced distance travelled (Fig. 9c) and increased latency in the centre compared with littermate controls

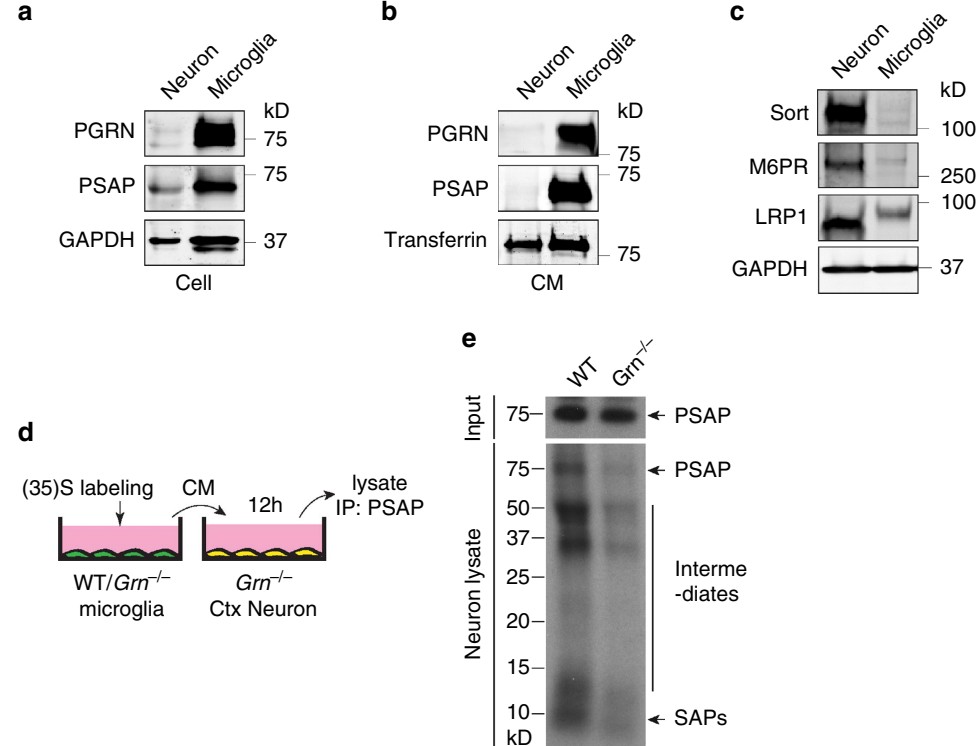

**Figure 5 | Microglial PGRN facilitates neuronal uptake of PSAP.** (**a**) Lysates from WT microglia and DIV12 cortical neurons were immunoblotted with anti-PGRN, anti-PSAP and anti-GAPDH antibodies. (**b**) Conditioned media (CM) from WT microglia and DIV12 neurons were immunoprecipitated with anti-PGRN, anti-PSAP or anti-transferrin antibodies and the IP products were subject to SDS–polyacrylamide gel electrophoresis (SDS-PAGE) and immunoblotting with corresponding antibodies. (**c**) Lysates from WT microglia and DIV12 cortical neurons were immunoblotted with anti-sortilin, anti-M6PR, anti-LRP1 and anti-GAPDH antibodies. (**a**-**c**) are representative blots from two independent experiments. (**d**-**f**) Application of radiolabelled medium from WT microglia results in more PSAP uptake and processing in neurons than medium from $Grn^{-/-}$ microglia. WT and $Grn^{-/-}$ primary microglia were labelled with $^{35}$S-labelled methionine and $^{35}$S-labelled cysteine. Labelled media were applied to DIV12 cortical neurons from $Grn^{-/-}$ mice. After 12 h, neuronal lysates were prepared and subject to anti-PSAP immunoprecipitations (**d**). The immunoprecipitates were separated using tricine gels and visualized by radiography (**e**). Representative neuronal uptake of full-length PSAP in $Grn^{-/-}$ neurons and WT controls.

(Fig. 9d), indicating disinhibition-like behaviour. Similar to *Pgrn* heterozygous knockout mice[32], sociability is affected in $Psap^{+/-}$ mice as they spent significantly less time with a stranger than control mice in the social interaction test (Fig. 9e). In addition, the number of direct interactions with the stranger mouse is also reduced in $Psap^{+/-}$ mice compared with controls, without any significant changes in olfactory sensitivity (Fig. 9f and data not shown). Next, we tested for short-term memory using the novel object recognition test. $Psap^{+/-}$ mice have a reduced preference score for a novel object compared with controls, showing that short-term memory is impaired (Fig. 9g). Interestingly, $Psap^{+/-}$ has no obvious effect on motor function in the rotarod test, similar to what is seen in *Pgrn*-deficient mice (Fig. 9h). Thus, PSAP haploinsufficiency in mice also leads to FTLD-related behavioural changes.

## Discussion
Despite accumulating evidence supporting a crucial role of PGRN in maintaining proper lysosomal function, how PGRN does so remains unclear. In this study, we demonstrate that PGRN determines the levels of neuronal saposins by facilitating neuronal uptake and lysosomal trafficking of PSAP through sortilin and that loss of saposin function contributes to FTLD-like phenotypes in mice (Fig. 10). Our data provide insight into how PGRN regulates lysosomal function and suggest a novel disease mechanism for subsets of FTLD and NCL due to *GRN* mutations.

Saposins are essential for proper glycosphingolipid degradation in the lysosome[21–23]. The saposin precursor PSAP can be delivered to lysosomes from the biosynthetic pathway or from the extracellular space. Since a large portion of PSAP is secreted, lysosomal delivery of PSAP from the extracellular space may be a significant source of lysosomal saposins in neurons, especially under glia activation, when the levels of PSAP are upregulated in glial cells (microglia and astrocytes) and most PSAP are secreted from glial cells (Figs 4,5a,b and Supplementary Fig. 2). PSAP strongly interacts with PGRN both within the cell and in the extracellular space[19]. Our studies with primary cortical neurons reveal two pathways for neuronal PSAP uptake, through LRP1 or through piggyback ride from PGRN and the PGRN receptor sortilin (Fig. 3). Given the strong dependence of neuronal PSAP levels on PGRN in the aged human brain, we speculate that the PGRN receptor, sortilin, might have a critical role in lysosomal delivery of PGRN-PSAP in such conditions. Unfortunately, the cell surface levels of trafficking receptors, which determine the endocytic trafficking routes for the PGRN–PSAP complex, are difficult to measure *in vivo*. Another possibility is that as PGRN levels increase in microglia during ageing, PGRN has a more critical role in neuronal uptake of PSAP via sortilin-mediated endocytosis.

In our previous work, we have demonstrated that PSAP facilitates lysosomal trafficking of PGRN via M6PR and LRP1[19]. Consistent with our study, PSAP was shown to regulate plasma PGRN levels in humans[33]. Thus, by forming a complex,

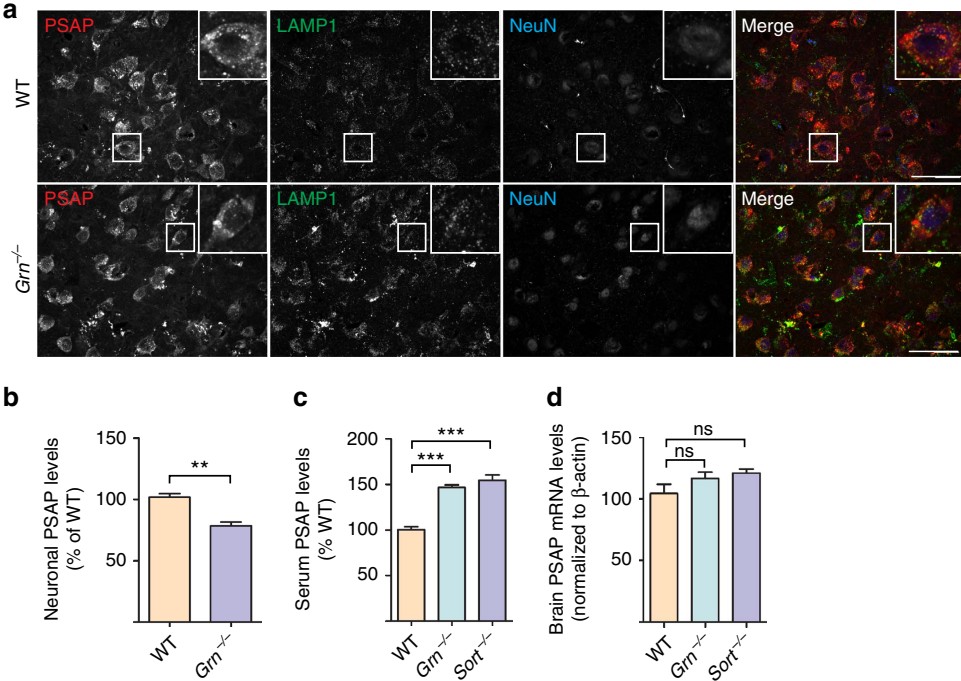

**Figure 6 | PGRN- and sortilin-deficient mice display PSAP trafficking defects. (a)** Representative images from immunostaining of 12–14-month-old brain sections of WT and $Grn^{-/-}$ mice with anti-mouse PSAP, LAMP1 and NeuN antibodies. Scale bar, 50 µm. **(b)** Quantification of immunofluorescence intensity of PSAP in neurons for images in **a**, $n = 4$, **$P < 0.01$, Student's $t$-test. **(c)** ELISA to measure PSAP levels in the serum of 6–8-month-old WT, $Grn^{-/-}$ and $Sort^{-/-}$ mice; $n = 5$, ***$P < 0.001$, one-way analysis of variance (ANOVA). **(d)** Quantitative PCR (qPCR) analysis of PSAP mRNA levels in the brain of 6–8-month-old WT, $Grn^{-/-}$ and $Sort^{-/-}$ mice; $n = 3$, NS, not significant, one-way ANOVA. Data are presented as mean ± SEM.

PSAP and PGRN facilitate each other's lysosomal trafficking. Distinct trafficking mechanisms might be used by different cell types under certain conditions, depending on the availabilities of the three trafficking receptors involved.

Although this study focuses on the regulation of PSAP by PGRN in neurons, another interesting observation in the current study is that glial cells, including microglia and astrocytes, upregulates PGRN and PSAP levels on injury and inflammatory conditions, such as traumatic brain injury and during FTLD and AD disease progression, and during normal ageing (Figs 4,7 and Supplementary Fig. 2). Whether PSAP–PGRN interaction regulates glial activation and whether that is related to the lysosomal functions of PSAP and PGRN remain to be explored. A role of PGRN in regulating lysosomal activity and complement activation in microglia has been demonstrated recently[34]. It will be interesting to test whether the PSAP–PGRN interaction has a role in this.

In addition to facilitating PSAP lysosomal trafficking, PGRN may regulate other aspects of PSAP biology, such as PSAP processing, and PGRN may have other functions in lysosomes in addition to regulating PSAP. Proper lysosomal function is essential for long-term neuronal survival[35,36]. Emerging studies suggest an interesting link between early-onset lysosomal storage diseases[37] and late-onset neurodegenerative diseases. The *GRN* gene is one example in which complete loss of PGRN causes NCL and haploinsufficiency of PGRN leads to FTLD. Another example is the glucocerebrosidase (*GBA*) gene. While loss-of-function mutations in *GBA* cause Gaucher disease, a lysosomal storage disorder, heterozygous mutation of *GBA* is associated with Parkinson's disease[38,39]. These observations suggest that a generalized impairment of lysosomal function may be a key driver of pathogenesis in late-onset neurodegenerative diseases, including FTLD. Besides *GRN*, other FTLD-associated genes,

*VCP/p97*, *CHMP2B*, *SQSTM1*, *TBK1* and *OPTN*, are involved in membrane trafficking and the autophagy–lysosome pathway[40].

In summary, our study demonstrates a role of the FTLD protein, PGRN, in determining neuronal saposin levels by facilitating lysosomal trafficking of PSAP and argues that impaired saposin function might be one of the disease mechanisms of FTLD-TDP with *GRN* mutations (Fig. 10). Our results underscore the importance of maintaining lysosomal saposin levels in preventing neurodegeneration in FTLD-TDP and give novel insights into the therapeutic development of FTLD-*GRN*.

## Methods

**Antibodies.** The following antibodies were used in this study: mouse anti-FLAG (M2) (1:2,000 for western blot), mouse anti-myc (9E10) (1:1,000 for western blot), rabbit anti-LRP1 (1:1,000 for western blot) from Sigma, mouse anti-GAPDH (1:5,000 for western blot) and rabbit anti-human PSAP (1:1,000 for western blot) antibodies from Proteintech Group, mouse anti-v5 (1:5,000 for western blot) from Invitrogen, sheep anti-mouse PGRN (1:1,000 for western blot, 1:100 for immunostaining), goat anti-human PGRN (1:1,000 for western blot, 1:100 for immunostaining) and goat anti-mouse sortilin (1:1,000 for western blot, 1:100 for immunostaining) from R&D systems, mouse anti-human LAMP1 (1:100 for immunostaining) and rat anti-mouse LAMP1 (1D4B) (1:100 for immunostaining) from BD Biosciences, mouse anti-GFAP (GA5) (1:100 for immunostaining) from Cell signaling, goat anti-cathepsin D (C20) (1:100 for immunostaining) from Santa Cruz, rabbit anti IBA-1 (1:250 for immunostaining) from Wako, mouse anti-mouse β-III tubulin (1:5,000 for western blot) from Promega, chicken anti-human transferrin (1:2,000 for western blot) from Immunology Consultants Laboratory, rabbit anti-p62/SQSTM1 (1:1,000 for western blot) from Novus, mouse anti-ubiquitin (1B3) (1:1,000 for western blot) from MBL, mouse anti-pTDP43 (S109/410) (1:1,000 for western blot) from COSMO Bio and mouse anti-NeuN (1:100 for immunostaining) from Millipore. Rabbit anti-mouse PSAP (1:100 for immunostaining) antibodies were characterized previously[19]. Rabbit anti-human PSAP (1:1,000 for western blot, 1:100 for immunostaining) and rat anti-human PSAP (1:1,000 for western blot, 1:100 for immunostaining) antibodies were generated by Pocono Rabbit Farm and Laboratory using the recombinant GST-PSAP proteins purified from bacteria. Validation of both antibodies is shown in Supplementary Fig. 9. Goat anti-mouse

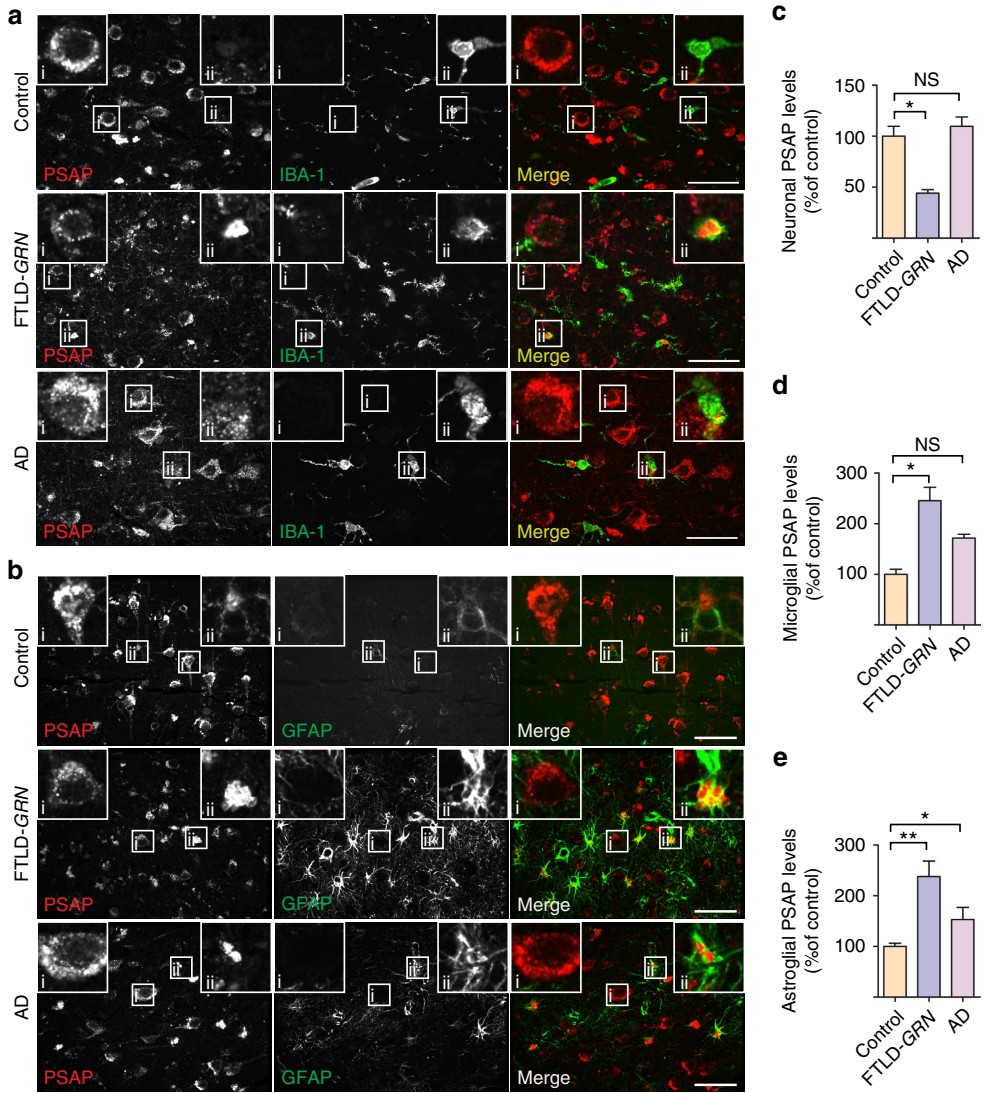

**Figure 7 | PSAP levels are increased in microglia and astrocytes but decreased in neurons in FTLD-*GRN* patients.** (**a**) Brain sections from control, FTLD-*GRN* and AD patients were stained with rat anti-PSAP and rabbit anti-IBA1 (marker for microglia) antibodies. A representative neuron was shown in inset i and a representative microglia was shown in inset ii. In contrast with increased PSAP levels in microglia, neuronal PSAP levels are much reduced in FTLD-*GRN* cases, but not in AD cases. Scale bar, 50 μm. (**b**) Brain sections from control, FTLD-*GRN* and AD patients were stained with rabbit anti-PSAP and mouse anti-GFAP (marker for astrocytes) antibodies. A representative astrocyte was shown in the inset. FTLD-*GRN* has many more activated astrocytes with high PSAP expression. Scale bar, 50 μm. (**c**) Quantification of neuronal PSAP levels in **a**; $n = 3$, *$P < 0.05$, NS, not significant, one-way analysis of variance (ANOVA). Data are presented as mean ± SEM. (**d**) Quantification of microglial PSAP levels in **a**; $n = 3$, *$P < 0.05$, NS, not significant, one-way ANOVA. Data are presented as mean ± SEM. (**e**) Quantification of astroglial PSAP levels in **b**; $n = 3$, *$P < 0.05$, **$P < 0.01$, one-way ANOVA. Data are presented as mean ± SEM.

PSAP (1:100 for immunostaining) antibodies[41] were a gift from Dr Ying Sun (Cincinnati Children's hospital, Cincinnati, OH). Rabbit anti-human saposin A, B, C and D (1:100 for immunostaining) antibodies[42] were a gift from Dr Xiaoyang Qi (University of Cincinnati School of Medicine, Cincinnati, OH). Rabbit anti-M6PR (1:1,000 for western blot) antibodies[43] were a gift from Dr William Brown (Cornell University, Ithaca, NY). Rabbit anti-SCMAS (1:100 for immunostaining) antibodies[44] were a gift from Dr Elizabeth F. Neufeld (David Geffen School of Medicine, University of California, Los Angeles, CA).

**Mouse strains.** C57/BL6 and *Grn*[-/-][45] mice were obtained from the Jackson Laboratory. Sortilin-knockout (*Sort*[-/-]) mice[46] were a gift from Dr S. Strittmatter (Yale University, New Haven, CT) and Dr A. Nykjaer (Aarhus University, Aarhus, Denmark). *Psap*[-/-] NA mice[30] and *Psap*[+/-] mice[47] were provided by Dr Ying Sun (University of Cincinnati). The age of the mice was described in each specific experiment. Both male and female mice were used and the gender of the mice in each experiment was matched in the same experiment. All the mice were housed in the

Weill Hall animal facility at Cornell. All animal procedures have been approved by the Institutional Animal Care and Use Committee (IACUC) at Cornell.

**DNA and plasmids.** His-tagged PSAP constructs were generated by cloning PSAP into pSetage2B vector (Invitrogen) using *Hind*III and *Not*I sites as described previously[19]. FLAG-tagged PSAP constructs were generated by inserting 3 × FLAG into His-PSAP using *Sfi*I and *Hind*III sites. AP-human PSAP construct was generated by cloning PSAP into pAP5 vector (Genhunter). Mouse PSAP-myc-his and human sortilin-myc-his constructs were generous gifts from Dr Carole Morales (McGill University). Human sortilin in a mammalian expression vector was obtained from Origene. Human PGRN in pCMV-Sport6 vector was obtained from Open Biosystems. GST-RAP construct is kindly provided by Dr Alban Gaultier from the University of Virginia.

**Cell culture.** HEK293T, Neuro2a (N2a) and COS-7 cells (ATCC) were maintained in Dulbecco's modified Eagle's medium (CellGro) supplemented with 10% fetal bovine serum (Gibco) and 1% penicillin–streptomycin (Invitrogen) in a humidified

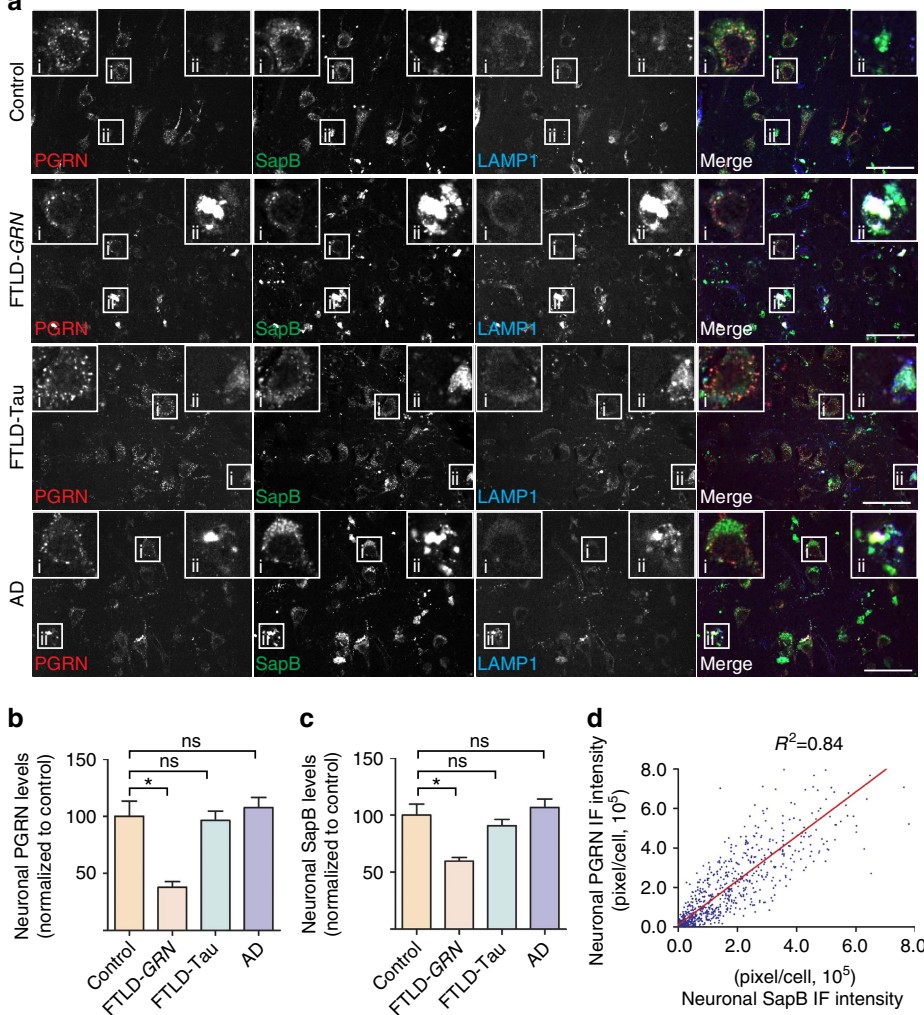

**Figure 8 | PGRN haploinsufficiency results in reduced neuronal saposin B levels in patients with FTLD.** (**a**) Brain sections from controls and patients with FTLD-TDP due to *GRN* mutations, FTLD-tau and AD patients were stained with anti-PGRN, anti-saposin B and anti-LAMP1 antibodies. Scale bar, 50 μm. A representative neuron is shown in inset i and a representative glia cell is shown in inset ii. (**b,c**) Quantification of neuronal PGRN and PSAP signals in controls, FTLD-TDP due to *GRN* mutations, FTLD-tau and AD for experiment in **a** using Image J; $n = 3$, $^*P < 0.05$; NS, not significant; one-way analysis of variance (ANOVA). Data are presented as mean ± SEM. (**d**) Correlation between immunostaining intensity of neuronal PGRN and saposin B for experiment in **a**. Total of 703 neurons from controls, FTLD-TDP with *GRN* mutations, FTLD-tau and AD were quantified.

incubator at 37 °C and 5% $CO_2$. Cells were transiently transfected with polyethyleneamine as described[48]. N2a cells with PGRN deletion were generated by infecting N2a cells with lentivirus expressing Cas9 only or Cas9 and guide RNA targeted to mouse *Grn* (oligos with 5′-CACCGCGGACCCCGACGCAGGT AGG-3′ and 5′-AAACCCTACCTGCGTCGGGGTCCGC-3′ were ligated to pLenti-CRISPR (Addgene)). Cells were selected with puromycin 2 days after infection and the knockout is confirmed by western blot and immunostaining.

Primary cortical neurons were isolated from P0 to P1 pups using a modified protocol[49]. Briefly, cortices were rapidly dissected in 2 ml Hanks' balanced salt solution supplemented with B27 (Invitrogen) and 0.5 mM L-glutamine (Invitrogen) at 4 °C. After removing meninges, the cortices were digested with papain (Worthington LS003119, 2 mg ml$^{-1}$ in Hanks' balanced salt solution) and DNaseI (1 mg ml$^{-1}$ in Hanks' balanced salt solution; Sigma) for 12 min at 37 °C. Fire-polished glass pipettes were then used to dissociate the tissues. Cells were spun down and resuspended in Neuroplex medium (Gemini) plus B27 and plated onto poly-lysine-coated dishes (Sigma). Mitotic inhibitors cytarabine (AraC; Sigma) were added at DIV3.

Microglia were isolated from P0 to P2 pups and grown on astrocytes for 2 weeks before being shaken off according to a published protocol[50].

**Cell surface-binding assay.** AP-PSAP-binding assays as described previously[18]. Briefly, conditioned media (CM) with AP or AP-PSAP generated from HEK293T cells were incubated with sortilin-transfected COS-7 cells or for 2 h at room temperature or primary cultured cortical neurons 1 h at 4 °C before fixation and heat inactivation of endogenous AP at 65 °C overnight. Bound AP to the primary cortical neurons was measured using the NIH image software.

**Protein preparation and western blot analysis.** Purification of recombinant his-human PSAP and his-human PGRN proteins, co-IP assays and western blots were performed as described previously[19]. GST-RAP is purified from BL21 cells using GST beads. To analyse p62, pTDP43 and ubiquitin in mouse brain, the frozen brain tissue were homogenized and lysed in ice-cold RIPA buffer (50 mM Tris, pH 7.3, 150 mM NaCl, 1% Triton, 0.1% SDS, 0.5% deoxycholic acid, 1 mM EDTA) with protease and phosphatase inhibitors (Roche Complete Mini, EDTA-free protease inhibitor, 10 μM MG-132, 5 mM NaF and 10 mM β-glycerol-phosphate). RIPA-soluble fraction was collected from the supernatant of cell lysis after a centrifugation at 18,000 g for 20 min at 4 °C. The RIPA-insoluble pellets were extracted with urea buffer (7 M urea, 4% CHAPS, 30 mM Tris, pH 8.5) and centrifuged at 18,000 g for 20 min at 4 °C and the supernatant was saved as the RIPA-insoluble, urea-soluble fraction. Full size western blots are shown in Supplementary Fig. 10.

**Metabolic labelling and PSAP processing assay.** To obtain the $^{35}$S-labelled mouse PSAP, the HEK293T cells were transfected with mouse PSAP-myc-his overnight. The medium was replaced with methionine- and cysteine-free DMEM with 10% dialysed FBS for 2 h before the addition of $^{35}$S-labelled methionine and cysteine. After 12 h incubation, the [$^{35}$S] isotope containing medium was replaced with serum-free medium for another 12 h, and then conditional media were collected. Sortilin-overexpressed COS-7 cells were then treated with $^{35}$S-labelled PSAP containing medium in the presence or absence of PGRN for 12 h. COS-7 cells were lysed with lysis buffer (50 mM Tris, pH 7.3, 150 mM NaCl, 1% Triton X-100 and 0.1% DOC with protease inhibitors). After

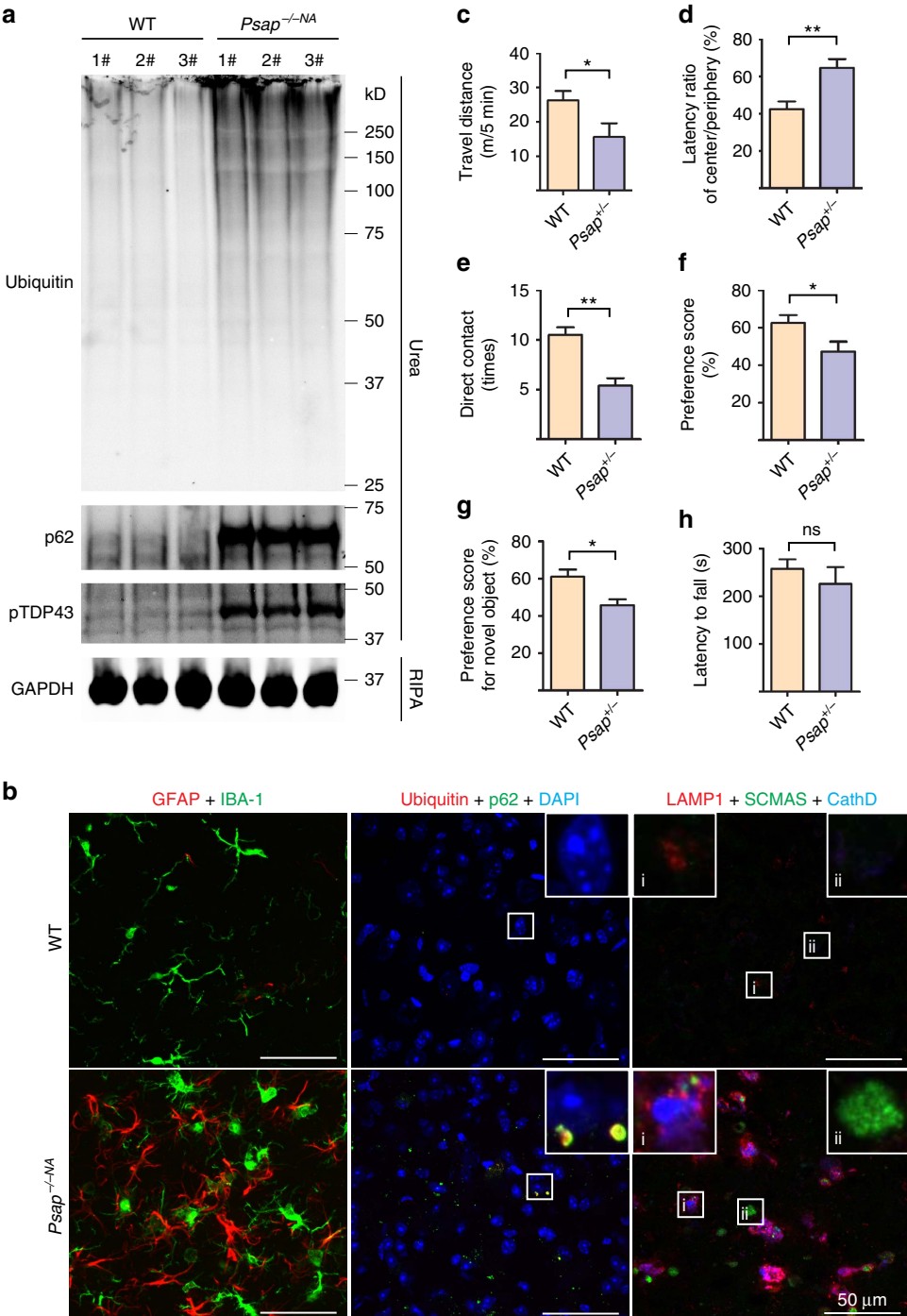

**Figure 9 | Partial loss of PSAP leads to FTLD-like phenotypes in mice. (a)** RIPA- and urea-soluble fractions from 6-month-old WT and $Psap^{-/-NA}$ brain were blotted with anti-ubiquitin, p62, phospho409/410 TDP-43 and GAPDH antibodies as indicated. **(b)** Brain sections from 6-month-old WT and $Psap^{-/-NA}$ mice were stained with anti-GFAP, anti-IBA1, anti-ubiquitin, anti-p62 or anti-SCMAS, anti-LAMP1 and anti-CathD antibodies as indicated. Scale bar, 50 μm. A representative glia cell with high LAMP1 levels is shown in inset i and a representative neuron is shown in inset ii for anti-SCMAS staining. The representative images from three mouse brains were shown. **(c,d)** Twelve-month-old WT and $Psap^{+/-}$ mice were subject to open-field test and $Psap^{+/-}$ mice shows significant reduction in travel distance and increase in latency in the centre arena. **(e,f)** $Psap^{+/-}$ mice show deficits in the sociability test. Data are presented as mean ± SEM. **(g)** $Psap^{+/-}$ mice show deficits in the novel object test. **(h)** $Psap^{+/-}$ mice do not show any significant motor deficits in the rotarod test. For all the behavioural tests, WT, $n=7$; $Psap^{+/-}$, $n=6$. *$P<0.05$; **$P<0.01$; NS, not significant.

immunoprecipitation with rabbit anti-mouse PSAP antibody, the IP products were separated with 18% tricine gels. Followed by fixation solution (10% methanol and 10% acetic acid), the gels were subsequently impregnated with amplify solution (1 M sodium salicylate, 10% glycerol) and the autoradiographs of dried gels were obtained on an X-ray film at −80 °C. To assess the neuronal uptake and processing of microglial PSAP, $^{35}$S-labelled mouse PSAP containing medium

were generated from primary cultured microglia, and then applied to primary cultured cortical neurons.

**PCR with reverse transcription.** Mouse cortical tissue was dissected and frozen in liquid nitrogen. Total RNAs were extracted using TRIzol (Invitrogen) and purified with Quick RNA MiniPrep Kit (Zymo Research). One microgram of total

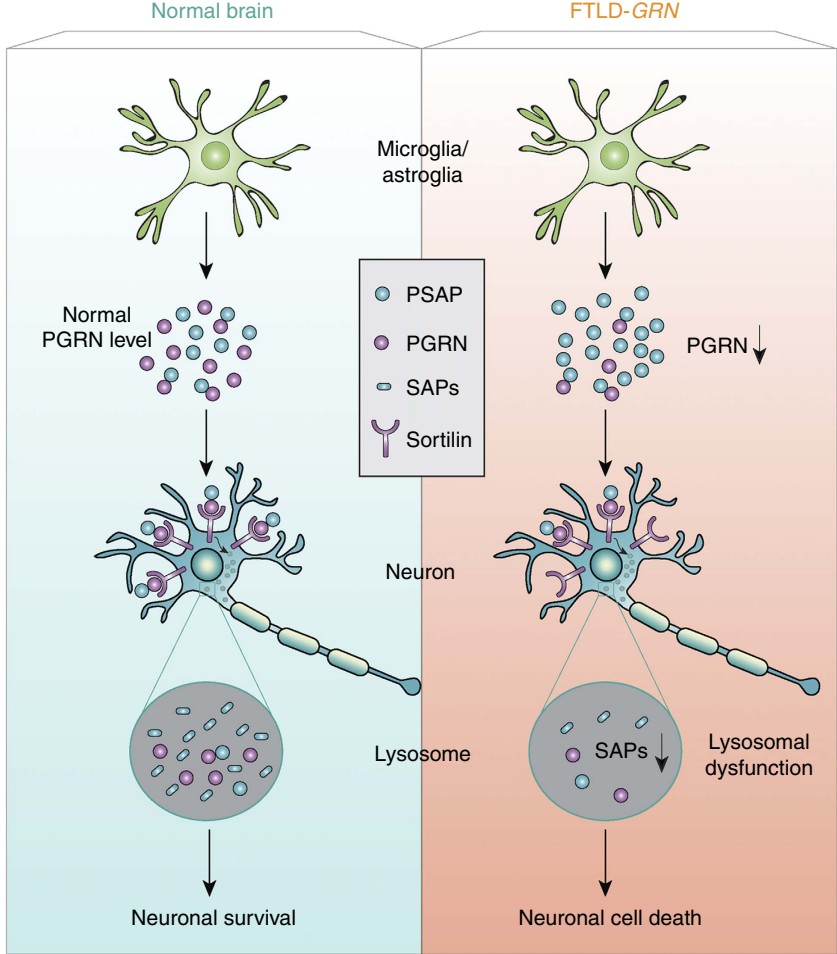

**Figure 10 | A schematic drawing illustrating the proposed disease mechanism of FTLD with *GRN* mutations.** PGRN and PSAP are highly secreted by microglia and astrocytes. Through binding to sortilin on neuronal cell surface, PGRN facilitates neuronal uptake of extracellular PSAP. Lysosomal delivery of PSAP results in PSAP processing into individual saposins (SAPs), which helps maintain normal lysosomal function in neurons. PGRN mutations in FTLD results in reduced PGRN levels and thus less neuronal uptake of PSAP and reduced saposin levels in neuronal lysosomes, which leads to lysosomal dysfunction and eventually neuronal cell death and FTLD. PSAP receptors, LRP1 and M6PR, which mediates alternate pathways for PSAP lysosomal delivery, are not shown in the drawing.

RNA was reverse transcribed to cDNA using poly (T) primer and SuperScript III Reverse Transcriptase (Invitrogen). Quantitative PCR was performed on a Light-Cyler 480 (Roch Applied Science), and the transcripts levels were measured using efficiency-adjusted $\Delta\Delta - CT$. PSAP transcript was normalized to β-actin. The mouse PSAP primer pair sequences were 5′-CCTGTCCAAGACCCGAAGAC-3′ and 5′-CAAGGAAGGGATTTCGCTGTG-3′. Mouse β-actin primers were 5′-ACGAGGCCCAGAGCAAGAG-3′ and 5′-TCTCCAAGTCGTCCCAGTTG-3′.

**Enzyme-linked immunosorbent assay.** To measure PSAP levels, serum samples were collected from WT, $Pgrn^{-/-}$, $Sort^{-/-}$ and $Psap^{-/-}$ mice. Diluted mouse serum (1:10) and purified mouse PSAP (100, 50, 25, 12.5, 6.25, 3.1, 1.5 and 0 ng ml$^{-1}$) were coated onto 96-well ELISA plates (NUNC; Thermo Scientific) at 4 °C overnight. The plates were then blocked with 1% bovine serum albumin in PBS at room temperature for 1 h after extensive washes. Plates were then washed and incubated with home-made rabbit anti-mouse PSAP antibodies[19] (1:4,000) 4 °C overnight. After several washes, plates were then incubated with horseradish peroxidase-conjugated goat anti-rabbit antibodies (Vector Laboratories) at room temperature for 30 min. The plates were washed four times and incubated with solution C mixed from solution A and B (ABC Kit, Vector Laboratories) at room temperature for 30 min in the dark. After washing, the plates were incubated with chromogenic substrate, 3,3′,5,5′-tetramethylbenzidine (TMB), for 15 min. The reaction was stopped by adding 2 M H$_2$SO$_4$. The plates were read at 450 nm (real signal) and 540 nm (background). Three repeats were tested for each sample. The specificity of the antibody was confirmed using serum from PSAP$^{-/-}$ mice as controls (Supplementary Fig. 9).

**Brain tissue.** Human brain tissues were obtained from the Neurodegenerative Disease Brain Bank at the University of California, San Francisco. Authorization for autopsy was provided by patients' next-of-kin, and procedures were approved by the UCSF Committee on Human Research. Neuropathological diagnoses were made in accordance with consensus diagnostic criteria[51,52]. Cases were selected based on neuropathological diagnosis and genetic analysis. Freshly frozen blocks and formalin-fixed, paraffin-embedded tissue sections of the anterior orbital gyrus were used from subjects with FTLD-TDP, Type A, due to *GRN* mutations, corticobasal degeneration (a subtype of FTLD-tau), AD and healthy controls. Healthy control tissue was obtained from individuals without dementia who had minimal age-related neurodegenerative changes. Detailed information is provided in Supplementary Table 1.

**Cortical stab wound injury.** Adult C57/B6 mice were anesthetized with isoflurane and placed in a stereotaxic apparatus. A 27 1/2-gauge needle was inserted 2.0 mm in depth and left in place for 1 min. The skin was then sutured, and the mice were allowed to recover for 4 days before killing.

**Immunofluorescence staining and quantitative analysis.** Immunofluorescence staining was performed as described previously[19]. For paraffin-embedded human brain samples, 8 μm sections were deparaffinized with xylene and ethanol. Antigen retrieval was performed by microwaving in citrate buffer (pH 6.0) for 18 min. To block the lipofuscin autofluorescence, brain sections were incubated with 0.1% Sudan Black B (Spectrum Chemical) in 70% ethanol for 20 min at room temperature before the staining process. Images were acquired on a CSU-X spinning disc confocal microscope (Intelligent Imaging Innovations) with an HQ2 CCD camera (Photometrics) using a ×40 and ×100 objective.

The quantitative analysis of the images was performed in ImageJ program. For the quantitative analysis of intracellular levels of PSAP, PGRN and SapB, the entire

cell body was selected and the fluorescence intensity was measured directly with ImageJ after a threshold application. For the quantitative analysis of lysosomal PSAP levels, the regions of interest for lysosomal areas were selected based on LAMP1 signals, the regions of interest were then applied to PSAP channel and the fluorescence intensity was measured. For each sample, —four to seven different random images, acquired using a $\times 40$ objective, were captured, and data from $\geq 3$ brains were used for statistical analysis.

**Behavioural test.** Twelve-month-old female $Psap^{+/-}$ mice and WT littermate controls in the C57/BL6 background are subject to the following behavioural tests: (1) open-field test was conducted in an arena of 30 cm $\times$ 30 cm $\times$ 30 cm. The total track distance, centre track distance, centre time and centre entries were tracked by the Viewer III software (Biobserve, Bonn, Germany). (2) For the novel object recognition test, mice were first exposed to two identical objects for 10 min, followed by a 2 h retention interval, and then put back to the same chamber where a novel object is introduced and monitored for 3 min. Exploration, defined as any type of physical contact with an object (whisking, sniffing, rearing on or touching the object), was recorded and analysed using the tracing software Viewer III and corrected manually. The preference score (%) for novel object was calculated as (exploration time of the novel object/exploration time of both objects) $\times$ 100%. (3) Social interaction test was performed in a three-chambered apparatus as described previously[53]. A test mouse was first placed in the middle chamber and allowed to explore the arena for 5 min. An unfamiliar mouse (stranger mouse) was then introduced into one of the empty wire cages. Then, door was reopened and the test mouse was allowed to freely explore both the empty chamber and the chamber containing the stranger mouse for 10 min. The time of the test mouse spent sniffing each wire cage was measured. The stranger mice used in this experiment were age-matched C57/BL6 female mice, not littermates. (4) *Rotarod test*: The rotarod apparatus (Biological Research Apparatus, Varese, Italy) was used to measure motor coordination and balance. During the training period, each mouse was placed on the rotarod at a constant speed (16 r.p.m.) for a maximum of 180 s. Mice received three trials per day for 4 consecutive days after a steady baseline was attained. Mice were then subject to three trials at 4 to 40 r.p.m. accelerating speed levels. The latency to fall off the rotarod was recorded and analysed. All behaviour tests were performed by experienced experimenter, who was blinded regarding mouse genotypes.

**Statistical analysis.** The data were expressed as means ± s.e.m. One-way analysis of variance (ANOVA) followed by Bonferroni's multiple comparison test was used to compare statistical significance between multiple groups. The Student's *t*-test was used to compare two groups. All statistical analyses were performed using the GraphPad Prism5 software (GraphPad Software). *P*-values $< 0.05$ were considered statistically significant.

**Data availability.** All data generated or analysed during this study are included in this published article (and its Supplementary Information files) or available from the authors on request.

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

## Acknowledgements

We thank Xiaochun Wu for technical assistance, Dr Ying Sun (University of Cincinnati) for goat anti-PSAP antibodies, $Psap^{-/-}$ and $Psap^{-/-NA}$ mice, Dr Stephen Strittmatter (Yale University, New Haven, CT) and Dr Anders Nykjaer (Aarhus University, Aarhus, Denmark) for $Sort^{-/-}$ mice, Dr Xiaoyang Qi (University of Cincinnati) for anti-human saposin antibodies, Dr Elizabeth F. Neufeld (University of California, Los Angeles, CA) for anti-SCMAS antibodies, Dr William Brown (Cornell University) for anti M6PR antibodies, Dr Carol Morales (Mcgill University) for PSAP and sortilin-expressing constructs and Dr Alban Gaultier (University of Virginia) for GST-RAP construct. Human tissue samples were provided by the Neurodegenerative Disease Brain Bank at the University of California, San Francisco, which receives funding support from NIH Grants P01AG019724 and P50AG023501, the Consortium for Frontotemporal Dementia Research and the Tau Consortium. This work is supported by funding to F.H. from the Weill Institute for Cell and Molecular Biology, the Alzheimer's Association, the Association of Frontotemporal Dementia (AFTD), the Muscular Dystrophy Association and NINDS (R21 NS081357-01, R01NS088448-01) and by funding to X.Z. from the Weill Institute Fleming Postdoctoral Fellowship.

## Author contributions

F.H. and X.Z. designed the study and analysed the data. X.Z. performed all the experiments except Fig. 1a–c and behavioural tests. F.H. characterized the physical interaction between PGRN, PSAP and sortilin. L.S. purified the recombinant proteins and helped with mouse work. J.W.C. and Y.J. helped with genotyping and mouse work. O.A.B. helped with COS-7 uptake assay. O.B. performed behavioural tests and J.C.C.H. performed stab injury under the supervision of N.N. A.L.N. and W.S. selected and provided human brain tissue samples and helped interpret the human data. F.H. supervised the entire project and wrote the manuscript together with X.Z. W.S., A.L.N., O.B. and O.A.B. edited the manuscript.
