## [Peer Review File · Nature Communications]

Reviewers' comments:

Reviewer #1 (Remarks to the Author):

PGRN is a secreted protein with multiple functions. Mutation of one copy of the gene for PGRN leads to frontotemporal lobar degeneration (FTLD) that is associated with accumulation of ubiquitinated inclusions that taine for TDP-43 (TAR DNA binding protein-43). Mutation of both copies leads to a lysosomal storage disorder, neuronal ceroid lipofuscinosis. Both conditons are associated with clearance disorders, whether the failure of ubiquitination to clear TDP43 from the cells, or lysosome disorder. The group Of Dr Hu has made important progress identifying the binding of PGRN to proteins associated with lysosomes or lysosme trafficking, namely sortilin and prosaposin. Here they extend this to show that PGRN markedly increases the uptake of prosaposin from the medium to the lysosomes, and that this ours via the formation of a sortilin-PGRN-prosaposin complex. They show that PGRN and PSAP levels increase in microglia following injury but decrease in neurons, and similarly in aptietns with FTLD due to PGRN mutationt here is also increase glial but decreased neuronal PSAP content. They suggest, plausibly, that PGRN secretion by microglia increases the uptake of PSAP by cortical neurons, and that mice lacking PGRN or sortilin have a defect in neuronal PSAP uptake. PSAP^{-/-} mice have behavioral defects similar to those expected from FTLD but lack the presence of TDP43 inclusions.

Comments: The work is very well done.

The uptake of progranulin by sortilin can be blocked using C-terminal PGRN peptides. It mght be worth establishing how well these block the uptake of prosaposin.

Fig 2c:PSAP plus PGRN lanes. It looks as though there are several vesicles with psap fluorecence but not PGRN. Is this correct or am I mis-interpreting the signals?

Figure 5. The figure is said to show that neuronal PSAP is lower in PGRN^{-/-} mice than WT. The data in the immunofluorescence panels (5a) is not that strong, and the quantitation (5b) shows a rather small decrease in neuronal PSAP. The serum data (5c) is more convinving. A different interpretation of 5a might be that in the absence of PGRN the PSAP is mislocated, as the overlap with LAMP1 seems rather incomplete. It might be worth including other markers, for example for the Golgi.

Fig 8b. What cell types are showing the ubiquitin and p63 staining?

Discussion. Here PSAP partial knockouts show behavioral defects similar to FTLD but no motor defects. In humans there is at least one case where mutation of PSAP lead to motor defects in infancy (J. Inherit. Metab. Dis. 23: 63-76, 2000). Several mutations of PSAP lead to leukodystrophy. This is not fully in line with the phenotypes described here. How would sortilin knockout behavioral defects compare to PSAP phenotypes described here? Although the evidence suggests the PSAP partial knockout mice were trending towards either TDP-43-proteinopathy or lipofuscinosis, they clearly did not have either strong full evidence of either of these phenotypes. Would the authors like to comment.

Reviewer #2 (Remarks to the Author):

This is an interesting set of findings that contributes novel insights into the role of PRGN in brain. The data provide a cohesive and reasonably well documented model showing how PGRN release from glia, including microglia and possibly reactive astrocytes, facilitates the neuronal uptake of PSAP by serving as a structural bridge to sortilin. The findings clarify the basis for the observed weak direct interaction between PSAP and sortilin. The in vitro studies are amplified with data from human FTLD brain and a mouse model of PSAP deficiency, with appropriate controls. The data explain the observed PSAP trafficking deficit resulting in decreased delivery of PSAP to neuronal lysosomes. One set of somewhat puzzling observation is, on the one hand, the correlation between PGRN levels and neuronal saposin B levels, and the fact that neuronal saposin D levels increase in

FTLD neurons that are deficient in PSAP, reflecting likely accumulation due to lysosomal dysfunction. This raises the questions of how much is the PGRN deficiency of saposin D contributing to the dysfunction of lysosomes (are there other more important factors) and are there relationships between the saposins themselves that regulate their levels in lysosomes? The trafficking story holds together and is worthwhile despite these open questions and the additional question of how saposin changes translate into changes in functional activity of lysosomes.

Additional comments:

1. In Fig 5, the amount of neuronal PSAP is lowered modestly. It looks like the colocalization with LAMP may be more impaired than the absolute level, which would be supportive of the trafficking defect, especially if the identity of the non-LAMP compartment containing PSAP were identified. In any event, it would be useful to determine the degree of colocalization of the labels. Is the modest reduction in PSAP levels suggesting that the role of PGRN is a minor one in this system?
2. Why are levels of PSAP not evaluated in microglia/astrocytes in human FTLD but not in the PRGN null mouse to support the proposed relationship?
3. Fig 6. The increases of PSAP in microglia are not consistent in the two examples (insets) shown.

Reviewer #3 (Remarks to the Author):

In their manuscript, Zhou and colleagues investigate the functional interaction of prosaposin (PSAP) and progranulin (PGRN) in the etiology of FTLD in mouse models and patient brain specimens.

The present study is an extension of earlier work in which the same authors proposed that loss of PGRN from lysosomes is the primary cause of lysosomal deficits seen in FTLD. In their earlier model, PSAP serves as a guiding factor to direct PGRN to lysosomes, a trafficking route that is dependent on LRP1 and the mannose-6 phosphate receptor, but independent of the PGRN receptor sortilin (Zhou et al., PNAS 2015). In the present study, the authors come up with a completely different model in which absence of PSAP from lysosomes is the primary insult in FTLD. In this new model, PGRN merely acts as a docking site to facilitate endocytic uptake of glia-derived PSAP by sortilin, an endocytic pathway proposed to deliver PSAP to lysosomes in neurons.

General comment:

The new model proposed in this manuscript largely relies on descriptive studies using immunostainings of primary neurons and histological sections from PSAP heterozygous and PRGN knockout mice as well as from FTLD patients. Because many of these data are poorly analyzed and not replicated in more than a single individual, they fail to provide strong support for the claims made by the authors. Also, no experiments to dissect the contribution of intracellular versus endocytic sorting pathways to PGRN-dependent delivery of PSAP to lysosomes are shown. Finally, no experiments are provided to actually query the involvement of sortilin or alternative PGRN receptors in neuronal uptake and lysosomal targeting of PSAP. Such experiments could be done easily by including primary neurons or histological sections from sortilin KO mice in the present study.

Specific concerns:

1) Many of the immunostainings lack quantification (e.g., Fig.1a, Fig. S1, Fig. 3, Fig. 6., Fig. 8b). In instances where quantifications are provided, wild type controls are typically set to 100% without including standard deviation in the statistical analysis (e.g., Fig. 2e, 4f, 5b, 5c). This is clearly inappropriate. Also, for most experiments, biological replicates are as few as n=3.

2) No immunoreactive band for PSAP is seen in Fig. 2d. Thus, it is unclear how the quantifications in Fig. 2e were derived.

3) The increase in PGRN and PSAP levels in the acutely injured mouse brain shown by immunostainings (Fig. 3) should be confirmed by Western blot analysis.

4) The PGRN/PSAP uptake studies in Fig. 4d-f should also be done after knockdown of receptors implicated in PSAP uptake (e.g., LRP1, sortilin, mannose-6 phosphate receptors) to identify the main receptor pathway implicated in neuronal uptake of PSAP (dependent or independent of PGRN). Such mechanistic insights are clearly mandatory to provide a significant conceptual advance to the field.

5) The proposed decrease in PSAP uptake in neurons from PGRN mutant mice is not at all obvious from the images provided in Fig. 5a. Also, One-way ANOVA rather than Student's t test is the proper method to test statistical significance of the quantifications shown in Fig. 5c-d.

6) Histological sections from single individuals with AD or FTLD-PGRN are depicted in Fig. 6 and 7. Thus, it is unclear how representative these findings really are.

7) PSAP gives rise to four proteolytic processing products SapA, SapB, SapC, and SapD. In Fig. 7, a decrease in neuronal SapB levels in one FTLD-PGRN patient is used as an argument for impaired delivery of the precursor polypeptide PSAP to lysosomes in the absence of PGRN. However, in Fig. S4, an increase in lysosomal levels of yet another PSAP processing product SapD is shown in the FTLD-PGRN patient. How do the authors explain the increase SapD levels if the precursor PSAP is not delivered to lysosomes in FTLD-PGRN?

8) The experiments shown in Fig. 8 should be replicated in sortilin-deficient mice to test whether a similar increase in pathological markers (e.g., p62, pTDP43) is also seen in the absence of the endocytic receptor for PGRN.

Reviewer #1 (Remarks to the Author):

PGRN is a secreted protein with multiple functions. Mutation of one copy of the gene for PGRN leads to frontotemporal lobar degeneration (FTLD) that is associated with accumulation of ubiquitinated inclusions that taine for TDP-43 (TAR DNA binding protein-43). Mutation of both copies leads to a lysosomal storage disorder, neuronal ceroid lipofuscinosis. Both conditions are associated with clearance disorders, whether the failure of ubiquitination to clear TDP43 from the cells, or lysosome disorder. The group Of Dr Hu has made important progress identifying the binding of PGRN to proteins associated with lysosomes or lysosme trafficking, namely sortilin and prosaposin. Here they extend this to show that PGRN markedly increases the uptake of prosaposin from the medium to the lysosomes, and that this ours via the formation of a sortilin-PGRN-prosaposin complex. They show that PGRN and PSAP levels increase in microglia following injury but decrease in neurons, and similarly in aptietms with FTLD due to PGRN mutation here is also increase glial but decreased neuronal PSAP content. They suggest, plausibly, that PGRN secretion by microglia increases the uptake of PSAP by cortical neurons, and that mice lacking PGRN or sortilin have a defect in neuronal PSAP uptake. PSAP+/- mice have behavioral defects similar to those expected from FTLD but lack the presence of TDP43 inclusions.

Reply: We thank the reviewer for recognizing the merit of our study.

Comments: The work is very well done.

The uptake of progranulin by sortilin can be blocked using C-terminal PGRN peptides. It mght be worth establishing how well these block the uptake of prosaposin.

Reply: Thanks. We have done new experiment with neuronal uptake of the PGRN mutant lacking the sortilin binding site (PGRN Δ 3aa) and showed that this mutant protein cannot be uptaken by cortical neurons and cannot enhance prosaposin uptake (new Figure 3).

Fig 2c: PSAP plus PGRN lanes. It looks as though there are several vesicles with psap fluorescence but not PGRN. Is this correct or am I mis-interpreting the signals?

Reply: Thanks for noticing this. It might be due to two reasons. The first is due to antibody sensitivity difference which results in stronger PSAP signal than PGRN. If we increase the brightness of the PGRN channel, more colocalization can be seen. Secondly, it could also be due to the difference in the kinetics of processing and turnover of PGRN and PSAP during trafficking. The half-life of PGRN in the endolysosomal compartment is still unknown. It could be that PGRN is degraded faster than PSAP. Also, our PSAP antibodies not only recognize PSAP but also saposin peptides. As we have shown in our JCB paper, saposin peptides do not bind to PGRN, thus during trafficking as prosaposin getting processed into saposin peptides, PGRN signals and prosaposin/saposin signals might not colocalize as they traffic through the endolysosome compartments.

Figure 5. The figure is said to show that neuronal PSAP is lower in PGRN-/- mice than WT. The data in the immunofluorescence panels (5a) is not that strong, and the quantitation (5b) shows a rather small decrease in neuronal PSAP. The serum data (5c) is more convinving. A different interpretation of 5a

might be that in the absence of PGRN the PSAP is mislocated, as the overlap with LAMP1 seems rather incomplete. It might be worth including other markers, for example for the Golgi.

Reply: We've done the immunostaining and image analysis quantifying PSAP signal in LAMP1 positive compartments in neurons, and showed ~25% reduction of PSAP in neuronal lysosomes in PGRN^{-/-} mice (new Figure S4), which is similar to changes in total PSAP levels in neurons (new Figure 6). Based on our model, we think this 25% reduction of neuronal lysosomal PSAP is due to less neuronal uptake of PSAP from the extracellular space, as ablation of PGRN in N2A cells using CRISPR does not affect lysosomal trafficking of PSAP from the Golgi (new Figure S5).

Fig 8b. What cell types are showing the ubiquitin and p63 staining?

Reply: Based on the nuclear size difference of neuron and glia, we believe that ubiquitin and p62 aggregates are present in both neuron and glia cells. We've also done costaining using mouse anti-NeuN and rabbit anti-p62 antibodies. We observed p62 positive aggregates surrounding NeuN signals, which are likely neuronal aggregates since NeuN is in the nucleus. We've also seen scattered aggregates, which is not close to NeuN, which are likely to be aggregates in glial cells. Unfortunately, due to the availabilities of cell type markers and compatibilities with our ubiquitin and p62 antibodies, we were unable to complete the full analysis.

Discussion. Here PSAP partial knockouts show behavioral defects similar to FTLN but no motor defects. In humans there is at least one case where mutation of PSAP lead to motor defects in infancy (J. Inherit. Metab. Dis. 23: 63-76, 2000). Several mutations of PSAP lead to leukodystrophy. This is not fully in line with the phenotypes described here. How would sortilin knockout behavioral defects compare to PSAP phenotypes described here? Although the evidence suggests the PSAP partial knockout mice were trending towards either TDP-43-proteinopathy or lipofuscinosis, they clearly did not have either strong full evidence of either of these phenotypes. Would the authors like to comment.

Reply: We thank the reviewer for recognizing the discrepancies in phenotypes of PSAP^{+/-} mice and human PSAP mutant patients. It has been reported that PSAP^{+/-} mice do not have any obvious phenotypes and we think this is due to the remaining WT allele of PSAP. In many cases of PSAP deficiency in humans, both alleles of PSAP are mutated, resulting in loss of PSAP function or the function of individual saposin peptides.

Regarding the phenotypes of sort^{-/-} mice, we've looked at TDP43 aggregation and NCL phenotypes in 12 month old sortilin^{-/-} mouse brain. But we failed to detect any obvious TDP43 aggregation and NCL phenotypes in sortilin^{-/-} brain, compared to age matched WT controls. According to the changes in PSAP levels in the serum, we assume the loss of neuronal PSAP levels in sort^{-/-} mice are similar to in PGRN^{-/-} mice (~25%), and this amount of neuronal loss of PSAP is much less compared to that in PSAP^{-/-}NA mice (>60%) (Sun et al., Molecular Genetics and Metabolism, 2002). Therefore, it is not unexpected that sort^{-/-} mouse brain lacks the TDP43 aggregation and NCL phenotypes observed in PSAP^{-/-}NA mouse brain.

Reviewer #2 (Remarks to the Author):

This is an interesting set of findings that contributes novel insights into the role of PGRN in brain. The data provide a cohesive and reasonably well documented model showing how PGRN release from glia, including microglia and possibly reactive astrocytes, facilitates the neuronal uptake of PSAP by serving as a structural bridge to sortilin. The findings clarify the basis for the observed weak direct interaction between PSAP and sortilin. The in vitro studies are amplified with data from human FTLN brain and a mouse model of PSAP deficiency, with appropriate controls. The data explain the observed PSAP trafficking deficit resulting in decreased delivery of PSAP to neuronal lysosomes. One set of somewhat puzzling observation is, on the one hand, the correlation between PGRN levels and neuronal saposin B levels, and the fact that neuronal saposin D levels increase in FTLN neurons that are deficient in PSAP, reflecting likely accumulation due to lysosomal dysfunction. This raises the questions of how much is the PGRN deficiency of saposin D contributing to the dysfunction of lysosomes (are there other more important factors) and are there relationships between the saposins themselves that regulate their levels in lysosomes? The trafficking story holds together and is worthwhile despite these open questions and the additional question of how saposin changes translate into changes in functional activity of lysosomes.

Reply: We thank the reviewer for recognizing the strength of our manuscript. We agree that the differences we observe between different saposin species are very puzzling. But this is in agreement with many reported cases of NCL, in which saposin D were identified as one of the main components of lipofuscin. After careful examination, we've found saposin D levels are increased only in the neurons with enlarged lysosomes, whereas its levels in the neurons with relative normal lysosomal size are actually decreased (new Figure S7a), which is consistent with other saposins. This strongly argue that the accumulation of saposin D in neurons might be due to different physical properties of these saposins that render saposin D difficult to be degraded in the lysosome especially when the lysosomal functions are compromised.

Additional comments:

1. In Fig 5, the amount of neuronal PSAP is lowered modestly. It looks like the colocalization with LAMP may be more impaired than the absolute level, which would be supportive of the trafficking defect, especially if the identity of the non-LAMP compartment containing PSAP were identified. In any event, it would be useful to determine the degree of colocalization of the labels. Is the modest reduction in PSAP levels suggesting that the role of PGRN is a minor one in this system?

Reply: We've done the immunostaining and image analysis quantifying PSAP signal in LAMP1 positive compartments in neurons, which showed ~25% reduction of PSAP in neuronal lysosomes in PGRN^{-/-} mice (new Figure S4). The reduction of neuronal PSAP signal is correlated with an increase in serum levels of PSAP in PGRN^{-/-} mice. Since we've seen modest reduction of PSAP levels in PGRN^{-/-} mice but a more robust effect in FTLN-PGRN brain, this would suggest that PGRN-sortilin pathway might play a bigger role in PSAP trafficking in humans than in mice.

2. Why are levels of PSAP not evaluated in microglia/astrocytes in human FTLN but not in the PGRN null mouse to support the proposed relationship?

Reply: We've added new data showing increased PSAP levels in microglia and astrocytes in PGRN-/- mouse brains (new Figure S3).

3. Fig 6. *The increases of PSAP in microglia are not consistent in the two examples (insets) shown.*

Reply: We've stained more human patient samples and have quantified the levels of PSAP in neuron, microglia, and astroglia in controls, FTLN-PGRN, and AD (new Figure 7). In the updated representative images in Figure 7a, we've shown that PSAP levels are very low in microglia from controls, but elevated in FTLN-PGRN and AD patients due to microglia activation. This result is quantified in Figure 7c. In Figure 7b, PSAP levels are relatively lower in the control samples with minimal glial activation but gets upregulated in the activated astrocytes in FTLN-PGRN and AD patients. The number of activated astrocytes is greatly increased in FTLN-PGRN patients, though, and almost all of these cells express much higher levels of PSAP compared to control samples (quantified in Figure 7e).

Reviewer #3 (Remarks to the Author):

In their manuscript, Zhou and colleagues investigate the functional interaction of prosaposin (PSAP) and progranulin (PGRN) in the etiology of FTLN in mouse models and patient brain specimens.

The present study is an extension of earlier work in which the same authors proposed that loss of PGRN from lysosomes is the primary cause of lysosomal deficits seen in FTLN. In their earlier model, PSAP serves as a guiding factor to direct PGRN to lysosomes, a trafficking route that is dependent on LRP1 and the mannose-6 phosphate receptor, but independent of the PGRN receptor sortilin (Zhou et al., PNAS 2015). In the present study, the authors come up with a completely different model in which absence of PSAP from lysosomes is the primary insult in FTLN. In this new model, PGRN merely acts as a docking site to facilitate endocytic uptake of glia-derived PSAP by sortilin, an endocytic pathway proposed to deliver PSAP to lysosomes in neurons.

Reply: We thank the reviewer for the summary of our work. The model presented in this manuscript is not in contrast to our previous published paper showing PSAP facilitates PGRN lysosomal trafficking (Zhou et al., JCB 2015). In fact, with primary cortical neurons, we've shown that PGRN and PSAP facilitate each other's uptake and lysosomal delivery (Figure 2 and 3).

General comment:

The new model proposed in this manuscript largely relies on descriptive studies using immunostainings of primary neurons and histological sections from PSAP heterozygous and PGRN knockout mice as well as from FTLN patients. Because many of these data are poorly analyzed and not replicated in more than a single individual, they fail to provide strong support for the claims made by the authors. Also, no experiments to dissect the contribution of intracellular versus endocytic sorting pathways to PGRN-dependent delivery of PSAP to lysosomes are shown. Finally, no experiments are provided to actually query the involvement of sortilin or alternative PGRN receptors in neuronal uptake and lysosomal targeting of PSAP. Such experiments could be done easily by including primary neurons or histological sections from sortilin KO mice in the present study.

Reply: We disagree with the reviewer that the new model is based on descriptive studies using immunostaining:

- (1) We've shown that PGRN bridges the interaction between PSAP and sortilin using multiple assays including co-IP (Fig. 1a, 1b) and AP binding assay (Fig. 1c).
- (2) We've shown that the interaction between PSAP and PGRN/sortilin leads to PSAP uptake and lysosomal delivery using endocytosis assay (Fig. 1d) and radiolabeling (Fig. 1e).
- (3) Based on results from these in vitro systems, we've shown that PGRN facilitates PSAP binding and uptake in primary cortical neurons using AP-binding assay, endocytosis assay and Western blot analysis (Fig. 2).
- (4) In response to the reviewers' comments, we've dissected the receptors involved in PGRN and PSAP uptake in cortical neurons (Fig. 3). Consistent with our previous findings, we've shown that PGRN uptake is sortilin dependent and PSAP uptake is LRP1 dependent and the presence of both PSAP and PGRN facilitate each other's lysosomal trafficking.
- (5) To identify in vivo relevance of our work, we've shown that PGRN and PSAP are highly expressed and secreted by glial cells (Fig. 4 and 5a-b). Furthermore, using radiolabeled conditioned medium from WT and PGRN^{-/-} microglia, we showed that PGRN deficiency leads less neuronal uptake of PSAP (Fig. 5d-f).
- (6) To determine the in vivo significance of our work, we've shown that lysosomal PSAP levels in neurons are reduced in PGRN^{-/-} mice, which is correlated with ~50% increase in serum levels of PSAP (Fig. 6a, 6b, 6c).
- (7) To determine the disease relevance of our work, we've shown FTLN patients with PGRN mutations but AD and FTLN-Tau show reduced levels of saposins in neurons (Fig. 7, Fig. 8, Fig. S6, Fig. S7). And the levels of PGRN in neurons directly correlate with saposin levels (Fig. 8d). This is based on the quantification of over 700 neurons in more than 3 patient samples of each genotype.
- (8) Furthermore, we've shown reduced levels of PSAP result in FTLN like pathology (Fig.9), supporting that PSAP trafficking defect could be a disease mechanism for FTLN-PGRN.

In response to the reviewer's comments:

- (1) We've looked at PSAP trafficking in the endocytic pathway by comparing PSAP lysosomal trafficking in WT and PGRN^{-/-} N2A cells generated using CRISPR. PGRN loss does not lead to PSAP lysosomal trafficking defect in these cells, which express high levels of sortilin, arguing that it is likely that PGRN/sortilin does not play a major role in PSAP lysosomal trafficking in the biosynthetic pathway (new Fig. S5).
- (2) We've dissected the receptors involved in PGRN and PSAP uptake in cortical neurons (Fig. 3). Consistent with our previous findings, we've shown that PGRN uptake is sortilin dependent and PSAP uptake is LRP1 dependent and the presence of both PSAP and PGRN facilitate each other's lysosomal trafficking.
- (3) We've quantified PSAP levels in microglia and astrocytes in control, FTLN-PGRN and AD samples using additional patient samples and these results in presented in new Fig. 7.
- (4) Please see other detailed responses below.

Specific concerns:

1) Many of the immunostainings lack quantification (e.g., Fig. 1a, Fig. S1, Fig. 3, Fig. 6., Fig. 8b). In instances where quantifications are provided, wild type controls are typically set to 100% without including standard deviation in the statistical analysis (e.g., Fig. 2e, 4f, 5b, 5c). This is clearly inappropriate. Also, for most experiments, biological replicates are as few as n=3.

Reply: We understand the reviewer's concern on the importance of quantification and reproducibility. We've tried our best to quantify the results except when the results are very clear without quantification. For the experiments without quantification, it is repeated at least once independently and similar results were obtained. We have added these descriptions in the figure legend.

We've also redone the statistical analysis and added deviation for WT controls when possible. For some experiments, we have to normalize the experimental group to the control group in the same experiment since the absolute value between experiments are too variable (such as Western blot analysis, it is very hard to get similar absolute values for experiments repeated at different times). Although n=3 is on the low side, it is statistically accepted number for analysis and we've obtained statistically significant difference between groups.

2) No immunoreactive band for PSAP is seen in Fig. 2d. Thus, it is unclear how the quantifications in Fig. 2e were derived.

Reply: We apologize about the low quality of the representative blots. We've repeated the experiments and the new results are shown in the new Fig. 2d. The quantification in 2e are derived from 3 independent experiments done at 3 different times. At each experiment, the amount of PGRN/PSAP uptaken as a complex is normalized to PGRN or PSAP alone.

3) The increase in PGRN and PSAP levels in the acutely injured mouse brain shown by immunostainings (Fig. 3) should be confirmed by Western blot analysis.

Reply: We've added new experiments with n=5 showing that both PGRN and PSAP are significantly upregulated after traumatic brain injury (new Fig. 4a-c).

4) The PGRN/PSAP uptake studies in Fig. 4d-f should also be done after knockdown of receptors implicated in PSAP uptake (e.g., LRP1, sortilin, mannose-6 phosphate receptors) to identify the main receptor pathway implicated in neuronal uptake of PSAP (dependent or independent of PGRN). Such mechanistic insights are clearly mandatory to provide a significant conceptual advance to the field.

Reply: We've carried out new experiment with the recombinant PGRN Δ 3aa protein, which has the last 3 aa deleted and thus cannot bind to sortilin (Zheng et al, PloS One 2011) to show that PGRN uptake is mediated by sortilin. Since we have trouble optimizing the conditions for siRNA and CRISPR in cortical neurons, we decided to treat the neurons with recombinant GST-RAP protein, a known antagonist for LRP1 (Hiesberger et al, EMBO 1998), to show that LRP1 plays a role in PSAP uptake in neurons. These results are included in the new Figure 3.

5) *The proposed decrease in PSAP uptake in neurons from PGRN mutant mice is not at all obvious from the images provided in Fig. 5a. Also, One-way ANOVA rather than Student's t test is the proper method to test statistical significance of the quantifications shown in Fig. 5c-d.*

Reply: Thanks. We've redone the statistical analysis as suggested.

6) *Histological sections from single individuals with AD or FTLN-PGRN are depicted in Fig. 6 and 7. Thus, it is unclear how representative these findings really are.*

Reply: The results in old Figure 6 are representative images from at least 2 individuals of the same category. To address the reviewer's concern, we've stained more samples from each group and done statistical analysis. The new data is presented in the new Fig. 7. The images in the old Fig. 7a are representative images from 3-4 individuals of the same category, followed by quantifications in the old Fig. 7b-7d (new Fig. 8). Quantification from over 700 neurons shows a direct correlation between PGRN levels and saposin B levels in Fig. 7d (new Fig. 8d).

7) *PSAP gives rise to four proteolytic processing products SapA, SapB, SapC, and SapD. In Fig. 7, a decrease in neuronal SapB levels in one FTLN-PGRN patient is used as an argument for impaired delivery of the precursor polypeptide PSAP to lysosomes in the absence of PGRN. However, in Fig. S4, an increase in lysosomal levels of yet another PSAP processing product SapD is shown in the FTLN-PGRN patient. How do the authors explain the increase SapD levels if the precursor PSAP is not delivered to lysosomes in FTLN-PGRN?*

Reply: As mentioned in response to reviewer #2's comment, we agree that the differences we observe between different saposin species are very puzzling. But this is in agreement with many reported cases of NCL, in which saposin D were identified as one of the main components of lipofuscin. After careful examination, we've found saposin D levels are increased only in the neurons with enlarged lysosomes, whereas its levels in the neurons with relative normal lysosomal size are actually decreased (new Figure S6a), which is consistent with other saposins. This strongly argue that the accumulation of saposin D in neurons might be due to different physical properties of these saposins that render saposin D difficult to be degraded in the lysosome especially when the lysosomal functions are compromised.

8) *The experiments shown in Fig. 8 should be replicated in sortilin-deficient mice to test whether a similar increase in pathological markers (e.g., p62, pTDP43) is also seen in the absence of the endocytic receptor for PGRN.*

Reply: As mentioned in response to Reviewer #1, we've looked at TDP43 aggregation and NCL phenotypes in 12 month old sortilin^{-/-} mouse brain. But we failed to detect any obvious TDP43 aggregation and NCL phenotypes in sortilin^{-/-} brain, compared to age matched WT controls. Since the extent of lysosomal PSAP loss in sortilin^{-/-} brain is likely to be similar to PGRN^{-/-} (~25%), it is not unexpected that sortilin^{-/-} mice do not mimic the phenotypes of PSAP^{-/-}NA (>60% reduction of PSAP) (Sun et al., Molecular Genetics and Metabolism, 2002).

REVIEWERS' COMMENTS:

Reviewer #1 (Remarks to the Author):

The authors have performed new experiments to address the queries I raised in the first version of this paper. They have provided detailed and thoughtful responses to my original questions. I believe they have made a satisfactory rejoinder to everything that I asked in the first review.

Reviewer #2 (Remarks to the Author):

The authors have provided satisfactory responses, including new supportive data, which have addressed concerns in the initial review. The manuscript is considerably improved.

Reviewer #3 (Remarks to the Author):

This reviewer appreciates the authors' efforts in addressing major issues raised in the review. Most of these issues have been addressed successfully and have significantly improved the clarity of this study.